# NPF-$k$CT: A $k$-CENTER CLUSTERING SOLVER WITH NEURAL PROCESS FILTER FOR CONTINUOUS POMDP-BASED OBJECT SEARCH

## ABSTRACT

Efficiently searching for target objects in intricate environments poses a significant challenge for mobile robots, due to perception errors, limited field of view (FOV), and visual occlusion. These factors cause the problem to be partially observed. Therefore, we formulate the object-search task as a high-dimensional Partially Observable Markov Decision Process (POMDP) with hybrid (continuous and discrete) action spaces. We propose a novel sampling-based online POMDP solver named Neural Process Filtered $k$-Center Clustering Tree (NPF-$k$CT). The optimal action is selected using Monte Carlo Tree Search (MCTS) in conjunction with a neural process network to filter out ineffective primitive actions (i.e., basic robot operations), alongside $k$-center clustering hypersphere discretization to efficiently refine high-dimensional continuous sub-action spaces. Adhering to the hierarchical optimistic optimization (HOO) concept, we leverage an upper-confidence bound (UCB) on the action value function within the hypersphere with estimated diameters to guide the MCTS expansion. We extensively tested our approach in Gazebo simulations using Fetch and Stretch robots across diverse target-finding scenarios. Comparative results show higher success rates and faster target detection than baseline methods, with no additional computational cost. We also validated our method on a physical robot in an office environment. Project page: https://sites.google.com/view/npfkct.

## 1 INTRODUCTION

Object-searching tasks for mobile robots with manipulators in cluttered, partially known environments, such as warehouses or living rooms, pose significant challenges. In scenarios such as retrieving an item from a warehouse box or fetching a Coke bottle from a kitchen table, the target object is often occluded. Known as the "Mechanical Search" problem, these tasks typically rely on partial knowledge of the environment—such as a rough map with large furniture or appliances like shelves and tables—alongside many unknown small objects, e.g. mugs or snack boxes. To succeed, the robot must adapt its configurations, strategically remove obstacles, localize the target, and ultimately plan and execute the necessary grasp actions. In this paper, we assume the robot operates in a home environment with multiple rooms and workspaces (furniture surfaces) and has access to 3D point cloud and 2D occupancy grid maps, along with photos of the target object (Fig. 1).

Object search with mobile manipulators with on-board sensors relies on a comprehensive support system encompassing modules for object segmentation, object detection, pose estimation, task-level planning, and motion planning. Significant advancements have been made in these areas, particularly with advanced learning technologies. For instance, Shaban et al. (2017) introduces a highly efficient one-shot learning method leveraging a Fully Convolutional Network for pixel-level image segmentation. Other methods employ diverse networks, such as the Neural Radiance Field architecture, to address challenges related to optimal grasp pose generation Sóti et al. (2023) and manipulation planning Qureshi et al. (2020). This work focuses on task-level planning to effectively select primitive actions and achieve long-horizon goals for target object search in complex environments.

In this paper, we propose a novel POMDP framework and solver, NPF-$k$CT, for object search tasks using mobile robot manipulators with only onboard sensors. We train a neural process-based net-

work to score primitive actions and filter useless ones before planning. Filtered actions are grouped into hyperspheres via clustering, and MCTS is used to construct a belief tree for optimal action selection. An adaptive strategy refines action domains by creating smaller clusters based on particle limits, ensuring precise sampling. The selected action is applied to the robot, updating the belief with real-world observations. We analyze the method's convergence under specific assumptions and show it outperforms baseline approaches, achieving efficient target search within POMDPs.

## 2 RELATED WORK

**Mechanical Search Planning Methodologies:** As previously discussed, robots frequently encounter challenges in navigating clustered environments while searching for target objects. Numerous existing methodologies are closely linked to advancements in learning technologies, such as deep reinforcement learning (RL) Kurenkov et al. (2020), deep Q-learning Yang et al. (2020), and deep-geometric inference systems Huang et al. (2021). In Kurenkov et al. (2020), the authors propose a learning procedure using an asymmetric architecture, suboptimal teacher guidance, and mid-level representations to train deep RL agents for uncovering occluded objects. However, these methods are highly customized for specific environments, like shelves and boxes, limiting their applicability in more diverse settings. Another significant framework for object search is the POMDP formulation.

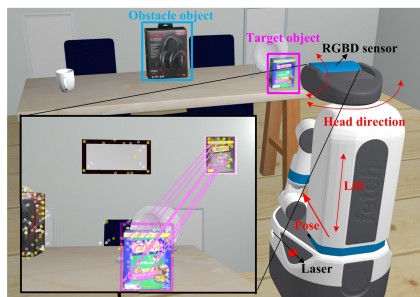

Figure 1: Fetch robot is pointed to search a snack box (pink). It adjusts its base, lift, and head (red) to change its FOV, with no prior knowledge (like size and number) of non-target objects.

In a notable example, as detailed in Huang et al. (2022), a MCTS method featuring a 1D occupancy distribution for objects is employed to swiftly identify and extract the target object. Due to the natural fitness of the different objects, the object-oriented POMDP (OO-POMDP) framework is widely applied to factor different objects, and the beliefs related to different objects are treated as independent due to the smaller computational cost Wandzel et al. (2019). As the demands of object-searching tasks evolve, the planning environment has progressed from a 2D plane scenario Aydemir et al. (2013) to a more complex 3D case Zheng et al. (2021). This transition incorporates a setting that accommodates different object dimensions using a multi-resolution planning strategy. Challenging the conventional object-independent assumption and acknowledging occlusion relationships, a recent work Chen & Kurniawati (2023) introduces an object-level POMDP formulation. This formulation entails a growing state space, incorporating a guessed target object, and is addressed through a novel solver based on MCTS and belief tree reuse, ultimately achieving a more efficient outcome.

**Mechanical Search Reasoning Methodologies:** In earlier robotics approaches, reasoning was often integrated into the planning process, such as updating probabilities, rather than being treated as a separate module. As robotic systems grew more complex, reasoning emerged as an independent layer, focusing on high-level cognitive tasks such as commonsense inference, contextual understanding, and hypothesis generation, which then guide a planning module to execute detailed action strategies. The work Giuliari et al. (2023) demonstrates a reasoning module to infer plausible object locations using environmental context and commonsense knowledge, aiding object localization in partially observed scenes. In more recent work Ge et al. (2024), the authors utilize commonsense knowledge from large language models to construct scene graphs, enhancing object search in household environments. However, these commonsense-based approaches struggle with unconventional object arrangements, like a random setting. In our scenario, objects are placed to violate human habit, increasing search complexity and necessitating active robot-environment interaction, such as removing occlusions. Our reasoning idea is involved in the probability update of the grid world. By representing the belief over the pose state of each object in the planning environment using particle filtering, the authors in Garrett et al. (2020) incorporate probabilistic reasoning into a deterministic planner and then complete the re-used replanning when facing the base movement failure. A recent work Huang et al. (2024) leverages a video tracking-based memory model with reasoning and planning capabilities, allowing the system to remember the potential locations of the occluded target objects and complete tasks by selecting action primitives.

**POMDP Formulation and Solvers:** For a POMDP agent, the process commences with a brief, where the agent deduces and executes the optimal action to transition to a new state. This new state is concealed within an updated belief, refined through Bayesian inference based on observed data. Following each step, the agent receives an immediate reward and contributes to a discounted cumulative reward, fostering a long-term objective. This iterative process persists until terminal conditions are met. This framework has found extensive application in the field of robotics, effectively addressing diverse tasks such as autonomous underwater vehicle navigation Hou et al. (2021), robot manipulation Pajarinen et al. (2022), and Human-Robot collaboration Burks et al. (2023). Attaining the exact optimal strategy for a POMDP problem is generally deemed computationally intractable Papadimitriou & Tsitsiklis (1987). Due to the cheaper memory demands compared with the offline solvers, the online methods, particularly sampling-based approaches, have emerged as predominant solutions, striking a favorable balance between approximate optimality and manageable computational load. Sampling-based solvers, like Partially Observable Monte Carlo Planning (POMCP) Silver & Veness (2010), Adaptive Belief Tree (ABT) Kurniawati & Yadav (2016), and Determinized Sparse Partially Observable Tree Somani et al. (2013), adopt a common approach of representing belief as particles and employ MCTS to expand the belief tree within constrained computational resources. Despite notable progress, addressing POMDPs with high-dimensional continuous action spaces remains a formidable challenge. The key point in existing continuous-action POMDPs is to refine the action subset incrementally to improve the possibility that the selected subset of actions contains the best action. Some continuous-action POMDP approaches, like partially observable Monte Carlo planning with observation widening (POMCPOW) Sunberg & Kochenderfer (2018), use the Progressive Widening (PW) strategy to continuously add new randomly sampled actions once current actions have been sufficiently explored. Other approaches incorporate technologies such as Voronoi Optimistic Optimization Lim et al. (2021) and Bayesian Optimization Mern et al. (2021) to enhance PW-based methods. These methods commonly employ the UCB1 Auer et al. (2002) algorithm for action selection during exploration and leverage Monte Carlo backup for value estimation.

**Neural Process in Robotics:** Neural processes (NPs) offer a powerful alternative to Gaussian processes (GPs) for function regression, capturing uncertainty as a stochastic process. In robotics, effectively managing prediction uncertainty is crucial for enhancing the robustness and applicability of systems in real-world scenarios. For example, Chen et al. (2022) introduces a meta-learning algorithm using Conditional Neural Processes (CNPs) to accurately estimate grasp points from depth images with minimal trials. Additionally, CNPs are utilized in a variety of applications, including 6D pose estimation Li et al. (2022) and social navigation Yildirim & Ugur (2022). In our work, we utilize neural processes to address the uncertainty in scoring function regression with high dimension input, like 3D point clouds and high-resolution images, and filter primitive actions effectively.

## 3 PROBLEM FORMULATION FOR OBJECT SEARCH

**General POMDP Model:** In this paper, we consider a POMDP $\mathcal{P}$ with a hybrid action domain. Formally, defined as an 8-tuple $< \mathcal{S},\ \mathcal{A},\ \mathcal{O},\ T,\ Z,\ R,\ \boldsymbol{b}_0,\ \gamma >$, where the state space $\mathcal{S}$ represents the state space, the action space $\mathcal{A} \triangleq \mathcal{A}_c \times \mathcal{A}_d$ denotes the set of all actions the robot can perform, where sub-domain $\mathcal{A}_c$ is assumed to be continuous and embedded in a bounded metric space with distance metric function and sub-domain $\mathcal{A}_d$ is discrete; the observation space $\mathcal{O}$ means the set of all observations the robot can perceive; the transition function $T(\boldsymbol{s},\ \boldsymbol{a},\ \boldsymbol{s}') = Pr(\boldsymbol{s}'|\boldsymbol{s},\ \boldsymbol{a})$ represents the nondeterministic effects of actions $\boldsymbol{a} \in \mathcal{A}$ working from the current state $\boldsymbol{s} \in \mathcal{S}$ to the resulting state $\boldsymbol{s}' \in \mathcal{S}$; the observation function $Z(\boldsymbol{s}',\ \boldsymbol{a},\ \boldsymbol{o}) = Pr(\boldsymbol{o}|\boldsymbol{s}',\ \boldsymbol{a})$ is commonly a conditional probability function that represents the observation the robot may perceive after performing action $\boldsymbol{a} \in \mathcal{A}$ in state $\boldsymbol{s}' \in \mathcal{S}$; the immediate reward function $R(\boldsymbol{s},\ \boldsymbol{a},\ \boldsymbol{s}')$ maps from a state, an action, a state–action pair, or a tuple of state, action, and subsequent state to a value; the state $\boldsymbol{s} \in \mathcal{S}$ is initially hidden in a initial belief $\boldsymbol{b}_0$, which is a probability distribution on the state space $\mathcal{S}$; $\gamma$ is a discount factor following $0 < \gamma < 1$, set as $0.9$ in this paper. The goal of solving a POMDP problem is to find an optimal policy $\Pi^*(\boldsymbol{b}) = \mathbf{argmax}_{\boldsymbol{a} \in \mathcal{A}} Q(\boldsymbol{b}, \boldsymbol{a})$ for belief $\boldsymbol{b}$, where the $Q(\boldsymbol{b}, \boldsymbol{a})$-value is the value of executing action $\boldsymbol{a}$ when the agent is at belief $\boldsymbol{b}$ and continuing optimally afterwards.

**Focused Object Search Problem:** Our task is to locate a target object in a complex environment containing multiple workspaces and numerous obstacles of unknown quantity and location, using a mobile robot equipped with onboard sensors. The state $\boldsymbol{s} \triangleq \{\boldsymbol{s}_r,\ \boldsymbol{s}_{o^0},\ \boldsymbol{s}_{o^1}, \cdots,\ \boldsymbol{s}_{o^n}\} \in \mathcal{S}$ comprises the robot's configuration and other objects, where $\boldsymbol{s}_r$ represents the robot state and $\boldsymbol{s}_{o^i}$

denotes the state of the $i$-th object. The robot state $\boldsymbol{s}_r = (\boldsymbol{p},\ l_h,\ l_p,\ l_t)$ includes the 6D base pose $\boldsymbol{p}$, lift height $l_h$, and the pan and tilt angles $l_p$ and $l_t$ of the robot's head[1]. At some stages, the robot may not detect any updating objects in the workspace. To guide the robot's actions, we introduce a guessed target object with state $\boldsymbol{s}_{o^0} = (\boldsymbol{p}_{o^0},\ s_{o^0}^x,\ s_{o^0}^y,\ s_{o^0}^z,\ \boldsymbol{g}_{o^0},\ m_{o^0},\ u_{o^0}) \in \mathbb{R}^{1\times 20}$, where $\boldsymbol{p}_{o^0} = (\boldsymbol{p}_{o^0}^p,\ \boldsymbol{p}_{o^0}^o)$ is the 6D object pose; $s_{o^0}^x$, $s_{o^0}^y$, and $s_{o^0}^z$ mean the sizes along the principal 3D axis of the object; $\boldsymbol{g}_{o^0} \in \mathbb{R}^{1\times 8}$ indicates 8 grid odds for identifying target/obstacle status, updated through object matching. The positions of these 8 grids are linked to the object's pose, indicating that each visual observation can only capture certain parts (surfaces or grids) of the object. This helps us to identify similar objects with the same visual surfaces. $m_{o^0} \in \mathbb{R}$ evaluates the object moveability, $u_{o^0}$ marks the object's status [2]. The belief of the position $\boldsymbol{p}_{o^0}^p$ of the guessed target object is saved in a grid world $\mathcal{G}_f$ generated by all workspaces $\mathcal{W}$ using many odds values $Odd(\mathcal{G}_f)$ with a given resolution. When sampling, within each grid cell, the object's position is uniformly sampled, and the probability is determined by the corresponding odds value[3]. These odds values $Odd(\mathcal{G}_f)$ are updated based on the camera FOV using real-world measurement in the excursion process, which is similar to the update of the occupancy grid map Chen et al. (2020); Zhao et al. (2024a). However, during belief tree search, the odds values $Odd(\mathcal{G}_f)$ remain unchanged and are used to sample the guessed target object for MCTS. The $i$-th detected objects after each real-world excursion will be appended to the state vector and form the sub-state $\boldsymbol{s}_{o^i}$ with the same variable format of $\boldsymbol{s}_{o^0}$.

Our control space $\mathcal{A}$ consists of 3 primitive actions: adjusting the robot's configuration (a continuous action in $\mathbb{R}^9$), declaring an object as a target or obstacle, and removing the $i$-th object using the robot manipulator. Since declaring and removing are discrete, the action space is a combination of continuous and discrete actions, $\mathcal{A}_c \times \mathcal{A}_d$. For simplicity, we assume all actions have a $100\%$ success rate[4]. The transition function $T$ mainly accounts for changes in camera motion due to the robot's configuration and changes in object status from declaring or removing actions. The camera motion follows the rigid transformation and the static structure information of the applied robot. Once an object is removed, it is moved outside the workspace, and its status $u_{o^i}$ is set to -2, indicating it will no longer block the view of other objects. The observation space $\mathcal{O}$ and function $Z$ focus on objects within the robot's head camera FOV frustum $V$. Observations are assumed to update the log-odds of the 4 nearest grids [5] within $\boldsymbol{g}_{o^i}$ corresponding to the observed objects with noise-adjusted values: $-c_o + \eta$ for negative log-odds, $c_o + \eta$ for positive, and $\eta$ for near-zero, where $c_o$ is a constant, and $\eta$ is Gaussian noise. If the original mean log-odds over 8 grids exceed a positive threshold $\nu_p$, the updated value will be $c_0 + \eta$. For values below a negative threshold $\nu_n$, the updated value will be $-c_0 + \eta$. Values between thresholds generate near-zero $\eta$. The discrete observation space is $\{\{i,\cdots,j\},\ \{\boldsymbol{o}(i),\cdots,\boldsymbol{o}(j)\}\}$, where $\{i,\cdots,j\}$ records observed objects within $V$, and $\{\boldsymbol{o}(i),\cdots,\boldsymbol{o}(j)\}$ are the updated log-odds value. Our reward function $R$ encourages declaration and removal actions to complete the object search task. Rewards are structured as: $R_{max}$ for successfully removing the target object, $R_{c_t}$ for correctly declaring the target, $R_{c_o}$ for correct obstacle declarations, and $R_{min}$ for each action, where $R_{max} \gg R_{c_t} > R_{c_o} \gg 0 > R_{min}$. The cost of the removal action is set as $2R_{min}$ due to its complexity. All illegal actions, such as colliding with occupied grids, incorrect declarations, or re-removal attempts, incur a large penalty ($R_{ill} \ll 0$).

**Perception and Implementation Support:** To simplify the scenario with limited resources—a 2D lidar, 3D RGBD camera, and 7-DOF robot arm on the Fetch robot [6]—we assume a pre-built point cloud and occupancy grid map, including furniture and some known objects, is available for planning. Other objects with unknown identities and poses are excluded. The robot's configuration changes are implemented using ROS interfaces: move_base for base movement, AMCL for navigation Quigley et al. (2009), ros_control for joint control Chitta et al. (2017), and MoveIt for

---

[1] This description is based on the Fetch robot, but our framework can be adapted to other mobile robots with similar configurations by minor revision.

[2] -2 indicates the object has been removed; -1 means it is still updating without being declared or removed; 0 and 1 signify it has been declared as an obstacle or target object, respectively, and is no longer updating.

[3] Note that the grids for workspaces differ from 8 grids associated with objects. Please refer to Appendix A.

[4] To ensure a 100% success rate, we use the Gazebo server's set_model_state function with added Gaussian noise after standard control operations, though improvements will be made in future work.

[5] This is manual setting to simulate the visual surface. Others are also fine, like based on the visible grids.

[6] The Stretch robot also has a camera and lidar, so the perception system can be directly transferred. Its manipulator has only 3 degrees of freedom, so we use the IKPy tool Manceron (2022) to compute inverse kinematics, integrating base motion to achieve the required grasp poses.

declaration and removal actions Şucan et al. (2012). The observation after each action is mainly to estimate object poses, sizes, detectors, and move-ability for all detected objects. Object poses and sizes are derived from point cloud operations, including iterative closest point (ICP, initialized by 2D lidar matching) Rusu & Cousins (2011), filtering, segmentation (Euclidean cluster extraction), and principal component analysis (PCA). Object detection is achieved through point cloud reprojection, sub-image fusion, YOLO Rusu & Cousins (2011), SIFT Ng & Henikoff (2003), and color matching to enhance robustness across different object types. Similarly, the object's move-ability is evaluated by virtually planning the possible trajectory using the learning-based Grasp Pose Detection (GPD) toolbox Ten Pas et al. (2017), the $k$-means clustering algorithm, and the ROS moveit toolbox. Since this paper does not focus on the implementation system, please refer to Appendix B for more details.

## 4  NPF-$k$CT SOLVER

NPF-$k$CT is an anytime online POMDP solver. We assume that the Q-value of the considered POMDP problem follows Lipschitz continuity in the action space. NPF-$k$CT follows a standard procedure with four alternating stages: planning, execution, obtaining observations, and filtering, as shown in Algorithm 1 and Fig. 13. It primarily focuses on the planning stage, aiming to identify the optimal action based on the current belief $b_0$. To reduce the complexity of the high-dimension action domain and improve the refining efficiency, a pre-trained neural process-based network predicts the feasibility of sampled actions with associated uncertainties, filtering out the useless action domains. The actions are then clustered into hyperspheres using $k$-center clustering. A belief tree $\mathcal{T}$ is constructed, where nodes represent beliefs and actions. Each belief node $node_o \in \mathcal{T}$ is linked to a dynamic list $\mathcal{L}(node_o)$, which is initially formed from the previous step's tree and then fuses with the newly generated hyperspheres, displaying all the connected action nodes. During episode simulations, $\mathcal{L}(node_o)$ expands as the action space is refined further using $k$-center clustering.

---

**Algorithm 1** NPF-$k$CT framework

---

**Input:** Initial belief $b_0$
**Output:** Task is complete or not
1: $b \leftarrow b_0$, isTerminal = False
2: **while** isTerminal is False **do**
3:    ——————Planning stage——————
4:    $\mathcal{A}_r = \{a_i\} \leftarrow$ Net_sam($Odd(\mathcal{G}_f), \mathcal{P}, b$))
5:    $\mathcal{C}_a = \{center_i\}$, $\mathcal{R}_a = \{range_i\} \leftarrow k$-Clustering($\mathcal{A}_r$)
6:    $\mathcal{L}(node_o) =$ Update($\mathcal{T}, \mathcal{C}_a, \mathcal{R}_a$)
7:    **while** planning budget not exceeded **do**
8:      $s \leftarrow$ Sampling($b$)
9:      $\mathcal{T} \leftarrow$ Episode_simulation($\mathcal{T}, s, h$)
10:    **end while**

11:    ——————Execution and filtering stage——————
12:    $a^*, center^*, range^* \leftarrow$ Get the best action in $\mathcal{T}$ from $b$
13:    **while** planning budget not exceeded **do**
14:      $a_{imp} \leftarrow$ Action_sam($a^*, center^*, range^*$)
15:      **if** Reasonablility_check($a_{imp}$) **then**
16:        break
17:      **end if**
18:    **end while**
19:    ($o$, isTerminal) $\leftarrow$ Execute($a_{imp}$)
20:    $b, \mathcal{T} \leftarrow$ Filtering($b, a_{imp}, o$)
21: **end while**

---

Algorithm 1 includes functions and parameters that are detailed below: Net_sam($\cdot$) is the function that generates small feasible regions by sampling numerous candidate actions $\mathcal{A}_r$ and filtering them using neural network (refer to Section 4.1); $k$-Clustering($\cdot$) is the function that generates high-dimension hyperspheres by partitioning and covering the candidate actions set using $k$-Clustering over some discrete actions, satisfying $|\mathcal{C}_a| = |\mathcal{R}_a| = k$; Update($\cdot$) is the function to expand the dynamic list of each belief node $\mathcal{L}(node_o)$ based on the filtered cluster centers and ranges; Episode_simulation($\cdot$) refers to the function that performs general MCTS sampling by refining the action domain for each particle (episode); Action_sam() is the function that selects a discrete action $a_{imp}$ from the chosen domain with center $center^*$ and radius $range^*$. For details on these belief tree-related functions, refer to Section 4.2. Reasonablility_check($\cdot$) ensures that only feasible discrete actions are selected, such as preventing a robot from moving into obstacles. Execute($\cdot$) executes the selected action on the platform and obtains real observations (see Section 3). Filtering($\cdot$) is the particle filter to get a new belief and its corresponding sub-tree.

### 4.1  NPF-$k$CT: NEURAL PROCESS NETWORK FOR FILTERING

**Motivation about Network Filtering:** In POMDPs, many primitives within the continuous sub-action domain $\mathcal{A}_c$ may be inefficient or even unreasonable for pursuing long-term goals or even short-term ones, like completing a primitive action. If the optimal action $a^* = \mathbf{argmax}_{a \in \mathcal{A}_c} Q(b, a)$

lies within $\mathcal{A}_c$, there exists a smaller but more efficient feasible region $\mathcal{X} \subset \mathcal{A}_c$ such that $\boldsymbol{a}^* \in \mathcal{X}$ and $Q(\boldsymbol{b}, \boldsymbol{a})$ is relatively large. An intuitive idea is, given $\boldsymbol{s}$ and $\alpha(\mathcal{P})$, to quickly identify this region $\mathcal{X}$ using a score function $g(\boldsymbol{a}, \boldsymbol{s}, \alpha(\mathcal{P})) > 0$, $\forall \boldsymbol{a} \in \mathcal{X}$, which, despite some uncertainty, can be computed efficiently using a Gaussian process (GP), where $\alpha(\mathcal{P})$ is all configuration settings of the problem $\mathcal{P}$. However, in high-dimensional POMDPs, GPs are computationally intensive and limited in applicability. Instead, we aim to use NPs, which are defined as distributions over functions, to estimate the uncertainty in their predictions. Rather than training a complex network for long-term goals, as seen in Q-learning Hausknecht & Stone (2015), with extremely large minimum description length Zhao et al. (2024b) to pursue $\mathcal{X} = \boldsymbol{a}^*$, we use a simpler scoring network focused on short-term (even one step), physically meaningful goals. This approach filters out ineffective actions, allowing a compact POMDP to select the optimal action based on current beliefs. Inheriting the advantage of the neural networks, NPs are computationally efficient during training and evaluation, and also flexible for different inputs. Hence, we use neural processes $nn(\mu(\boldsymbol{a}), \sigma(\boldsymbol{a}))$ to learn this scoring function, where $\mu(\boldsymbol{a})$ is the mean function and $\sigma(\boldsymbol{a})$ is the kernel function.

$$g(\boldsymbol{a}, \boldsymbol{s}, \alpha(\mathcal{P})) \sim nn(\mu(\boldsymbol{a}), \sigma(\boldsymbol{a})) \tag{1}$$

To preserve the optimal action $\boldsymbol{a}^*$, eliminate as many irrelevant actions as possible, and accurately represent the complex feasible region $\mathcal{X}$, we would like to get a set of $\{\boldsymbol{a}_i\} \subset \mathcal{A}_c$ such that with high probability, $g(\boldsymbol{a}, \boldsymbol{s}, \alpha(\mathcal{P})) \geq 0$ and then cover these samples using some high-dimension hyperspheres. We get a bound on the predictive scores of the samples:

**Theorem 1.** *Let* $g(\boldsymbol{a}, \boldsymbol{s}, \alpha(\mathcal{P})) \sim nn(\mu(\boldsymbol{a}), \sigma(\boldsymbol{a}))$, $\delta \in (0, 1)$ *and set* $\beta^* = (2 \log(1/\delta))^{\frac{1}{2}}$. *If* $\mu(\boldsymbol{a}_i) > \beta^* \sigma(\boldsymbol{a}_i)$, *all steps* $\forall i = 1, \cdots, T$, *then* $Pr[g(\boldsymbol{a}_i, \boldsymbol{s}, \alpha(\mathcal{P})) > 0, \forall i] \geq 1 - \delta$.

For the proof of this theorem, please refer to Appendix C. This theorem provides a condition for actions $\boldsymbol{a}_i$: If all the sampled actions using $nn(\mu(\boldsymbol{a}_i), \sigma(\boldsymbol{a}_i))$ satisfying $\mu(\boldsymbol{a}_i) > \beta_i^* \sigma(\boldsymbol{a}_i)$, then all samples will satisfy the constraint $g(\boldsymbol{a}_i, \boldsymbol{s}, \alpha(\mathcal{P})) > 0$ with probability at least $1 - \delta$. This simple conclusion offers a good way to sample good robot actions using their predictive scores.

**NPs filtering in object search tasks:** Based on state $\boldsymbol{s}$, the primary goal of the object search process is to observe target object (assumed to be $i$-th object) and then update the belief $\boldsymbol{b}(\boldsymbol{g}_{o^i})$ of the 8 grid odds value from an initial belief $\boldsymbol{b}_0(\boldsymbol{g}_{o^i})$ to the target belief $\boldsymbol{b}_T(\boldsymbol{g}_{o^i})$ by changing the robot's FOV. This process involves passing a certain threshold to enable subsequent declaration and removal actions. Among all primitive actions $\mathcal{A}$, those deemed *efficient* $\{\boldsymbol{a}_i, \cdots\} \in \mathcal{X}$ are identified if, at step $j$, they can update the belief $\boldsymbol{b}_j(\boldsymbol{g}_{o^i})$ to move closer to the target belief $\boldsymbol{b}_T(\boldsymbol{g}_{o^i})$ within a bounded distance: $\omega_\triangle \triangle \leq \|\boldsymbol{b}_{j-1}(\boldsymbol{g}_{o^i}) - \boldsymbol{b}_T(\boldsymbol{g}_{o^i})\|_1 - \|\boldsymbol{b}_j(\boldsymbol{g}_{o^i}) - \boldsymbol{b}_T(\boldsymbol{g}_{o^i})\|_1 \leq \triangle$, where $\triangle$ serves as a natural upper bound for grid updating, like updating one surface of the nontransparent object, since it is impossible to observe all surfaces of an opaque object simultaneously. $\omega_\triangle$, $0 < \omega_\triangle < 1$ means at least one grid is observed and updated correctly. According to Theorem 1, if the action $\boldsymbol{a}_i$ satisfies $\mu(\boldsymbol{a}_i) > \beta^* \sigma(\boldsymbol{a}_i)$, the probability that the grid belief moves closer to the target belief $\boldsymbol{b}_T(\boldsymbol{g}_{o^i})$ at step $j$ is at least $1 - \delta$. Given this, a successful action sequence that reaches the target belief and completes the task selected by the POMDP solver must include at least $N_l = [\|\boldsymbol{b}_0(\boldsymbol{g}_{o^i}) - \boldsymbol{b}_T(\boldsymbol{g}_{o^i})\|_1 / \triangle]$ efficient actions within $N_p \geq N_l$ potential primitive actions. Equality holds only if every primitive action is efficient. In order to complete the task faster with a better long-term reward, we had better get the actions with a larger $\beta^*$ satisfying $\mu(\boldsymbol{a}) > \beta^* \sigma(\boldsymbol{a})$. All these ideas rely on accurate learning of the scoring function. The scoring function $g(\boldsymbol{a}, \boldsymbol{s}, \alpha(\mathcal{P}))$ in our formulation is designed to learn the probability that the robot camera can observe the updated grids of the target object. This probability depends on the robot's configuration $\boldsymbol{s}_r$, obstacle data from the fused point cloud of detected objects $\{o_0, o_1, \cdots, o_n\} \in \mathcal{M}'_c$, the grid world status $\mathcal{G}_f$ (represented as a 2D grayscale image), and the 8 grid odds $\boldsymbol{g}_{o^i}$, satisfying $g(\boldsymbol{a}, \boldsymbol{s}, \alpha(\mathcal{P})) \triangleq g(\boldsymbol{s}_r, \mathcal{G}_f, \mathcal{M}'_c, \boldsymbol{g}_{o^i})$.

To learn the scoring function, we implement a repeating process to autonomously generate the simulation data using a Gazebo environment, shown in Appendix E. We first use some predefined action sequences to generate the point clouds about the detected objects. Then we keep changing the configuration of the robot and grid configuration, and use object detection to compute the probability of detecting the target object. The process is fully autonomous after offering given candidate actions and workspace $\mathcal{W}$. A scenario with a sampled target object, a given robot configuration, and a successful online color-based object detection is shown in Fig. 14.

For the network structure, we use various encoder networks: partial PointNet Qi et al. (2017), ResNet-18 He et al. (2016), and Multilayer Perceptron (MLP) to process different inputs. The

point cloud $\mathcal{M}'_c$ is encoded into a 1024-element global feature vector. The odds value for the grid world related to the guessed target object is converted into a 2D grayscale image and encoded as a 1000-dimensional vector using ResNet-18. The robot configuration and the status of the 8-grid odds are processed by the MLP, producing three 10-dimensional global features. These are concatenated into a global feature tensor of size $M \times 2054$. To use NPs, this tensor is divided into training data $M_1 \times 2054$ (including context data $(\boldsymbol{x_C}, \boldsymbol{y_C})$ and target data $(\boldsymbol{x_T}, \boldsymbol{y_T})$) and test data $M_2 \times 2054$, where $M_1 + M_2 = M$. The latent variable NP model then models the conditional distributions as: $p(\boldsymbol{y_T}|\boldsymbol{x_T}, \boldsymbol{x_C}, \boldsymbol{y_C}) := \int p(\boldsymbol{y_T}|\boldsymbol{x_T}, \boldsymbol{z})q(\boldsymbol{z}|rep(\boldsymbol{x_C}, \boldsymbol{y_C}))d\boldsymbol{z}$, where $rep(\boldsymbol{x_C}, \boldsymbol{y_C})$ is an encoder function that shows a representation of the context data using an MLP, $p(\star|\bullet)$ denotes the conditional prior for $\star$ given $\bullet$, and $q(\star|\bullet)$ means the variational posterior for $\star$ given $\bullet$. Then, with the latent variable $\boldsymbol{z}$ generated by the Gaussian sampling of the representation $rep(\boldsymbol{x_C}, \boldsymbol{y_C})$, the MLP-based decoder process is applied for the latent variable $\boldsymbol{z}$ and the test data to model a final Gaussian distribution for prediction. The whole network structure without training is shown in Appendix F. For training, the network about the encoder part to the latent variable needs to work on both the context data and the target data to get Kullback–Leibler divergence $D_{KL}(\bullet||\bullet)$ between prior and posterior. The parameters of the whole network are learned by maximizing the following evidence lower bound (ELBO) $\log p(\boldsymbol{y_T}|\boldsymbol{x_T}, \boldsymbol{x_C}, \boldsymbol{y_C}) \geq u_{ELBO}$:

$$u_{ELBO} = \mathbb{E}_{q(\boldsymbol{z}|\boldsymbol{x_C}, \boldsymbol{y_C})}\left[\log p(\boldsymbol{y_T}|\boldsymbol{x_T}, \boldsymbol{z})\right] + D_{KL}(q(\boldsymbol{z}|\boldsymbol{x_T}, \boldsymbol{y_T})||q(\boldsymbol{z}|\boldsymbol{x_C}, \boldsymbol{y_C})) \qquad (2)$$

Based on Le et al. (2018), the context data are selected as a subset of the target data, and the observation variance is learned as a latent variable within the range of 0.1 to 1 to enhance learning performance. Additionally, in the NP model, a self-attention is added to preprocess the context and test data tensors, which helps reduce predictive uncertainty near context points Kim et al. (2019).

## 4.2 NPF-$k$CT: BELIEF TREE CONSTRUCTION

**Construction overview:** To construct the belief tree $\mathcal{T}$, our NPF-$k$CT framework follows the typical select-expand-simulate-backup approach used in many MCTS algorithms. The key difference is the adaptive discretization using the $k$-center clustering method with a controllable discretization rate. As mentioned in the overview, each observation node has a dynamic list $\mathcal{L}(\boldsymbol{node_o})$. If resources permit, we continue sampling episodes to grow the belief tree. In each episode, a path is selected from the root: $Episode = \boldsymbol{s}_0, \boldsymbol{a}_0, \boldsymbol{o}_0, r_0, \boldsymbol{s}_1, \boldsymbol{a}_1, \boldsymbol{o}_1, r_1, \cdots$.

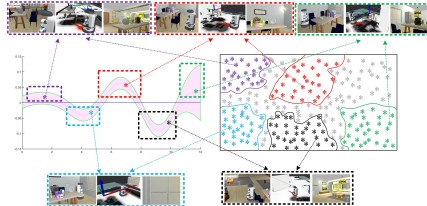

Figure 2: The NP-based scoring function and some scattered clusters.

Starting from the root belief, we first select an action $\boldsymbol{a}_i \in \mathcal{L}(\boldsymbol{node_o})$ using a revised UCB1 action selection strategy. We execute $\boldsymbol{a}_i$, moving from state $\boldsymbol{s}_i$ to $\boldsymbol{s}_{i+1}$, obtaining observation $\boldsymbol{o}_i$, reward $r_i$, and updating the belief from $\boldsymbol{b}_i = \{\boldsymbol{s}_i\}$ to $\boldsymbol{b}_{i+1} = \{\boldsymbol{s}_{i+1}\}$. If the process reaches a terminal condition or a child node does not exist, the tree is expanded by adding a new belief node with associated action nodes based on $\mathcal{L}(\boldsymbol{node_o})$. A rollout policy, typically a random action strategy Rollout_Random($\cdot$), is then simulated to estimate the value for the new node, followed by backup operations to update the values for visited nodes along the episode path. Please refer to Algorithm 2.

---

**Algorithm 2** Episode_simulation($\mathcal{T}, \boldsymbol{s}, \boldsymbol{h}$)

---

**Input:** The belief tree $\mathcal{T}$, sampled state $\boldsymbol{s}$, history in episode $\boldsymbol{h}$
**Output:** The discounted reward value $r$, the updated belief tree $\mathcal{T}$
1: Observation node $\boldsymbol{node_o} \leftarrow (N(\boldsymbol{node_o}), V(\boldsymbol{node_o}))$ based on history $\boldsymbol{h}$
2: **if** $\boldsymbol{node_a}$ is False **then**
3:    Get action nodes $\boldsymbol{node_a} \leftarrow (N(\boldsymbol{node_o}, \boldsymbol{a}), V(\boldsymbol{node_o}, \boldsymbol{a}))$ based on list $\mathcal{L}(\boldsymbol{node_o})$
4:    **return** Rollout_Random($\boldsymbol{s}, \boldsymbol{h} + \{\boldsymbol{a}, \boldsymbol{o}\}$)
5: **else**
6:    $\boldsymbol{a}^* = \text{argmax}_{\boldsymbol{a} \in \mathcal{L}(\boldsymbol{node_o})} U(\boldsymbol{node_o}, \boldsymbol{a})$
7:    $\mathcal{T} \leftarrow$ Refine($\mathcal{T}, \boldsymbol{node_o}, \boldsymbol{a}^*$)
8:    $\boldsymbol{a}_u^* \leftarrow$ Action_sam($\boldsymbol{a}^*, \boldsymbol{center}^*, range^*$)
9:    Get $\boldsymbol{s}'$ and $\boldsymbol{o}$ based on $\boldsymbol{a}_u^*$, $T(\boldsymbol{s}, \boldsymbol{a}^*, \boldsymbol{s}')$, and $Z(\boldsymbol{s}', \boldsymbol{a}^*, \boldsymbol{o})$
10:    **if** $\boldsymbol{s}$ is terminal state **then**
11:       $r \leftarrow \gamma$Episode_simulation($\mathcal{T}, \boldsymbol{s}', \boldsymbol{h} + \{\boldsymbol{a}^*, \boldsymbol{o}\}$) $+ R(\boldsymbol{s}, \boldsymbol{a}^*, \boldsymbol{s}')$
12:       Backup($\mathcal{T}, \boldsymbol{node_o}, \boldsymbol{a}^*, r, R(\boldsymbol{s}, \boldsymbol{a}^*, \boldsymbol{s}')$)
13:    **end if**
14: **end if**

---

**Action clustering and list initial update before MCTS:** Our key idea is to identify the feasible region $\mathcal{X}$ (Net_sam($\cdot$)) and update the action list associated with belief nodes (Update($\cdot$)). Due to

the complexity of real world problems, $\mathcal{X}$ is difficult to describe and often has a complex shape. A practical approach is to identify a set of potential actions $\mathcal{A}_r = \{\boldsymbol{a}_i\}$ with a high probability of satisfying $g(\boldsymbol{a}_i, \boldsymbol{s}, \alpha(\mathcal{P})) > 0$ using NPs filtering. The algorithm to obtain these samples is detailed in the Appendix G. The primary idea is to quickly obtain predicted mean and variance values with a single network evaluation and then test them based on Theorem 1. Typically, samples in the feasible region $\mathcal{X}$ form scattered clusters in the continuous action domain $\mathcal{A}_c$, as shown by the purple, red, and green regions in Fig. 2. We use the Elbow method Thorndike (1953) to determine the optimal number of clusters $k_i$ and all clusters are enclosed within high-dimensional hyperspheres with centers $\mathcal{C}_a$ and radii $\mathcal{R}_a$. The feasible region $\mathcal{X}$ is then a subset of the space covered by these hyperspheres, $\mathcal{X} \subseteq \bigcup_i \mathcal{H}(\boldsymbol{center}_i, range_i)$, $\boldsymbol{center}_i \in \mathcal{C}_a$, $range_i \in \mathcal{R}_a$. The centers and radii are recursively added to the action list $\mathcal{L}(\boldsymbol{node}_o)$ for observation nodes by traversing the entire tree.

**Action selection strategy and list growing in MCTS:** Inspired by the HOO idea Bubeck et al. (2011) from the continuous-arm bandit problem, we select an action from list $\mathcal{L}(\boldsymbol{node}_o)$ by:

$$\boldsymbol{a}^* = \mathrm{argmax}_{\boldsymbol{a} \in \mathcal{L}(\boldsymbol{node}_o)} U(\boldsymbol{node}_o, \boldsymbol{a}), \tag{3}$$

where $U(\boldsymbol{node}_o, \boldsymbol{a}) = \hat{Q}(\boldsymbol{node}_o, \boldsymbol{a}) + \omega_1 \sqrt{\frac{\log N(\boldsymbol{node}_o)}{N(\boldsymbol{node}_o, \boldsymbol{a})}} + \omega_2 range_i$, $\hat{Q}(\boldsymbol{node}_o, \boldsymbol{a})$ is the average rewards received in rounds when this action node was chosen; $U(\boldsymbol{node}_o, \boldsymbol{a})$ is the upper-confidence bound for the maximum possible Q-value in the hypersphere region $\mathcal{H}(\boldsymbol{center}_i, range_i)$, similar to the UCB1 bound. Our bound also considers the effect of $range_i$ for the $i$-th hypersphere following the Lipschitz assumption; $N(\boldsymbol{node}_o)$ and $N(\boldsymbol{node}_o, \boldsymbol{a})$ represent the number of visits to the observation node $\boldsymbol{node}_o$ and its corresponding action node, respectively; $\omega_1$ and $\omega_2$ are coefficients.

Within the planning budget, the episodes keep running from the root node and the action selection strategy in equation 3 is used for selecting the action node or expanding the belief tree $\mathcal{T}$. The action node will be refined and divided into several new clusters and hyperspheres, when an action node, which is associated with a high-dimensional hypersphere $\mathcal{H}(\boldsymbol{center}^*, range^*)$, is visited more than: $N(\boldsymbol{node}_o, \boldsymbol{a}^*) \geq 1/(C_r range^{*2})$. Here, $C_r$ is a self-defined exploration constant and $N(\boldsymbol{node}_o, \boldsymbol{a}^*)$ provides a rough estimate of the quality of the reward estimation $\hat{Q}(\boldsymbol{node}_o, \boldsymbol{a}^*)$, which follows the adaptive refining strategy in Hoerger et al. (2022) to limit the growth of the dynamic list $\mathcal{L}(\boldsymbol{node}_o)$ and ensures that a hypersphere is only refined when this action node has been visited sufficiently often. We also constrain the refining accuracy and limit the node number corresponding to $|\mathcal{L}(\boldsymbol{node}_o)|$ by $range^* \geq D_{lim}$, where $D_{lim}$ is the minimum radius for partitioning.

Assuming the action node $\boldsymbol{node}_{a^*}$ containing $N(\boldsymbol{node}_o, \boldsymbol{a}^*)$ episodes with the selected action $\boldsymbol{a}^*$ and hypersphere $\mathcal{H}(\boldsymbol{center}^*, range^*)$ needs refinement, these actions in this node are divided into $k$ clusters and then the corresponding hyperspheres are obtained with centers $\{\triangle\boldsymbol{center}_i\}$ and radii $\{\triangle range_i\}$, $i = 1, 2, \cdots, k$ based on the KMeans algorithm. We then update the action $\boldsymbol{a}^*$ of dynamic list $\mathcal{L}(\boldsymbol{node}_o)$ and its corresponding hypersphere $\mathcal{H}(\boldsymbol{center}^*, range^*)$ by the alternative actions set $\{\boldsymbol{a}^*, \boldsymbol{a}_{|\mathcal{L}(\boldsymbol{node}_o)|+2,}, \cdots, \boldsymbol{a}_{|\mathcal{L}(\boldsymbol{node}_o)|+k}\}$ and new hypersphere set $\{\mathcal{H}(\triangle\boldsymbol{center}_1, f_{lim}(\triangle range_1)), \cdots, \mathcal{H}(\triangle\boldsymbol{center}_k, f_{lim}(\triangle range_k))\}$, where $f_{lim}(\star)$ controls the refinement rate, ensuring convergence and planning performance.

$$f_{lim}(\star) = \max(f'_{lim}(\star), D_{lim}), \quad f'_{lim}(\star) = \begin{cases} \bar{\omega}_1 range^* & \text{if } \star \geq \bar{\omega}_1 range^* \\ \star & \text{if } \bar{\omega}_2 range^* < \star < \bar{\omega}_1 range^* \\ \bar{\omega}_2 range^* & \text{if } \star \leq \bar{\omega}_2 range^* \end{cases} \tag{4}$$

where $\bar{\omega}_1$ and $\bar{\omega}_2$ are coefficients controlling the refining velocity, with $0 \leq \bar{\omega}_2 < \bar{\omega}_1 \leq 1$. The original subtree with root node $\boldsymbol{node}_{a^*}$ is copied and connected to the observation node $\boldsymbol{node}_o$ as an additional child node based on actions $\boldsymbol{a}_{|\mathcal{L}(\boldsymbol{node}_o)|+2}, \cdots, \boldsymbol{a}_{|\mathcal{L}(\boldsymbol{node}_o)|+k}$. All nodes generated from $\boldsymbol{node}_{a^*}$ are updated based on the clustered episode IDs, shown in Algorithm 3 and Fig. 3.

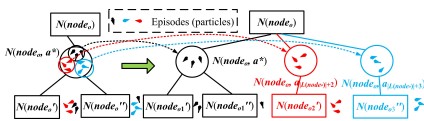

Figure 3: The refining process using clustered episode IDs.

**Action sampling and Backup:** Because the selected action $\boldsymbol{a}^*$ corresponds to the hypersphere $\mathcal{H}(\boldsymbol{center}^*, range^*)$ range, in order to execute the action in the POMDP problem, we assume that the sampled action is uniformly distributed in the hypersphere $\mathcal{H}(\boldsymbol{center}^*, range^*)$ and then sample a discrete action in this hypersphere, similar to ellipsoid sampling. When each episode reaches the terminal state, NPF-$k$CT framework updates the estimation reward $\hat{Q}(\boldsymbol{node}_o, \boldsymbol{a})$ as

---

**Algorithm 3** Refine($\mathcal{T}$, $node_o$, $a^*$)

---

**Input:** The belief tree $\mathcal{T}$, the observation node $node_o$, the selected action $a^*$
**Output:** The updated tree $\mathcal{T}$ with the refined nodes
1: Collects all applied actions $\mathcal{S}'_r$ in previous episodes passed leaf node of observation node $node_o$ with its action $a^*$
2: **if** $N(node_o, a^*) \geq 1/(C_r range^{*2})$ and $range^* > D_{lim}$ **then**
3:    $\{\triangle center_i\}$, $\{\triangle range_i\}$, clustered episode IDs $\leftarrow$ k-Clustering($\mathcal{S}'_r$)
4:    $\mathcal{L}(node_o) \leftarrow \mathcal{L}(node_o) \bigcup \{a_{|\mathcal{L}(node_o)|+2}, \cdots, a_{|\mathcal{L}(node_o)|+k}\}$
5:    $\mathcal{H}(center^*, f_{lim}(range^*)) \leftarrow \mathcal{H}(\triangle center_1, f_{lim}(\triangle range_1))$

6:    $\{\mathcal{H}(center_i, f_{lim}(range_i))\} \leftarrow \{\mathcal{H}(center_i, f_{lim}(range_i))\} \bigcup \{\mathcal{H}(\triangle center_2, f_{lim}(\triangle range_2)), \cdots, \mathcal{H}(\triangle center_k, f_{lim}(\triangle range_k))\}$
7:    Pick out sub-tree $\mathcal{T}_{sub}(node_o, a^*)$ using observation node $node_o$ and action $a^*$
8:    Copy and generate new sub-trees based on $\mathcal{T}_{sub}(node_o, a^*)$ and $\mathcal{L}(node_o)$
9:    Revise all its nodes based on clustered episode IDs, as shown in Fig. 3
10:    Attach generated sub-trees to $node_o$
11:    **return** updated tree $\mathcal{T}$
12: **end if**

---

well as visited numbers $N(node_o)$ and $N(node_o, a)$ of all nodes visited by this episode. Here, we present two classical stochastic backup methods including the Bellman backup, which is used in the ABT method and similar to the rule used in Q-learning, and the Monte-Carlo backup, which is widely used in many outstanding POMDP solvers, like POMCP, POMCPOW, and VOMCPOW. The detailed equations and some theoretical analysis of NPF-$k$CT are shown in Appendix H and I.

## 5 SIMULATION AND EXPERIMENTAL RESULTS

We validate our approach using Gazebo simulators and a real-world robot with C++ and Python. Evaluations span diverse object configurations, comparing our method to continuous-action benchmarks (POMCPOW and VOMCPOW) and classical POMDP methods with manually-set discrete actions (POMCP and GPOMCP). Additional settings are in Appendix J.

**Neural process for primitive action:** This part presents the prediction performance of the trained neural network in filtering meaningless primitives. Fig. 4 compares the observed test data with the predicted Gaussian distribution. The red line represents the observed data, and the predicted 2-$\sigma$ bound, truncated to [0, 1], is shown in pink. The prediction accuracy, $Acc = 1 - Pro_w$, where $Pro_w$ is the probability of misclassifying efficient actions (observed probability$>0.05$) as useless (mean $< 0.05$), is 99.02% for the test dataset. We also visualize results for two test samples. Overall, our neural network accurately filters out useless actions, enhancing the efficiency of the NPF-$k$CT solver by reducing the continuous action domain.

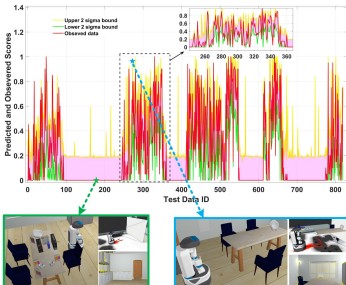

Figure 4: The NP results.

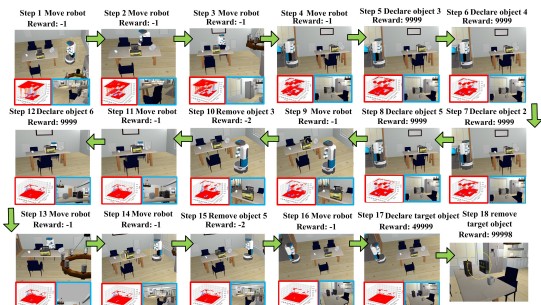

Figure 5: The visual progress for Covered1 scenario.

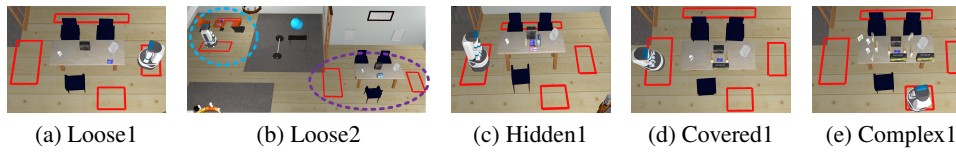

(a) Loose1        (b) Loose2        (c) Hidden1        (d) Covered1        (e) Complex1

Figure 6: The planning environment (red box: continuous action domain for robot position)

**Simulation Results:** Many object search methods rely on the classical POMCP approach with a discrete action domain. We compare our method with POMDP solvers in both discrete (POMCP, GPOMCP) and continuous action domains (POMCPOW, VOMCPOW). All methods were tested on

Table 1: Discounted cumulative reward, steps, and successful rate (within 50 steps)

| Scenarios | Loose1 | Complex1 |
|---|---|---|
| POMCP | $78629.6 \pm 8472.1 \mid 7.2 \pm 0.7 \mid$ **100**% | $27564.2 \pm 7591.0 \mid 47.9 \pm 3.2 \mid 25\%$ |
| GPOMCP | $79788.0 \pm 4787.4 \mid 7.4 \pm 0.6 \mid$ **100**% | $40278.0 \pm 4957.7 \mid 42.9 \pm 3.8 \mid 50\%$ |
| POMCPOW | $72612.0 \pm 11102.2 \mid 9.4 \pm 2.1 \mid$ **100**% | $37023.4 \pm 6951.9 \mid 44.2 \pm 3.4 \mid 75\%$ |
| VOMCPOW | $77622.9 \pm 11406.1 \mid 8.5 \pm 1.6 \mid$ **100**% | $40555.1 \pm 6830.9 \mid 41.1 \pm 4.4 \mid 90\%$ |
| NPF-$k$CT | **94795.1 $\pm$ 6350.6 $\mid$ 6.0 $\pm$ 0.7** $\mid$ **100**% | **44737.1 $\pm$ 6669.1 $\mid$ 36.3 $\pm$ 5.1 $\mid$ 95**% |

| Scenarios | Hidden1 | Covered1 |
|---|---|---|
| POMCP | $45815.2 \pm 7260.9 \mid 19.4 \pm 2.8 \mid$ **100**% | $31506.9 \pm 6249.7 \mid 32.3 \pm 6.0 \mid 80\%$ |
| GPOMCP | $55574.8 \pm 6225.3 \mid 15.3 \pm 2.0 \mid$ **100**% | $34397.8 \pm 7381.6 \mid 26.7 \pm 4.8 \mid 95\%$ |
| POMCPOW | $61728.4 \pm 8791.3 \mid 12.9 \pm 2.7 \mid$ **100**% | $40762.1 \pm 8401.1 \mid 23.9 \pm 6.3 \mid 90\%$ |
| VOMCPOW | $58286.8 \pm 10101.1 \mid 14.9 \pm 3.1 \mid$ **100**% | $35725.4 \pm 9880.5 \mid 25.2 \pm 5.9 \mid 90\%$ |
| NPF-$k$CT | **83377.1 $\pm$ 6427.3 $\mid$ 8.5 $\pm$ 1.4** $\mid$ **100**% | **44966.1 $\pm$ 6340.2 $\mid$ 21.7 $\pm$ 3.0 $\mid$ 100**% |

| Scenarios | Loose2 | |
|---|---|---|
| POMCP | $38462.6 \pm 11221.2 \mid 23.2 \pm 6.6 \mid 95\%$ | |
| GPOMCP | $51574.1 \pm 17930.9 \mid 17.4 \pm 2.9 \mid$ **100**% | |
| POMCPOW | $21785.1 \pm 6783.1 \mid 32.7 \pm 6.5 \mid 70\%$ | |
| VOMCPOW | $26860.0 \pm 5779.8 \mid 28.2 \pm 6.0 \mid 85\%$ | |
| NPF-$k$CT | **69992.7 $\pm$ 8185.8 $\mid$ 11.4 $\pm$ 2.0 $\mid$ 100**% | 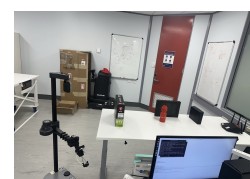 (b) Loose2 (Blue and purple dashed circles: two work areas) |

scenarios with varying object and workspace counts, including Loose1 (4 objects, 1 workspace), Loose2 (6 objects, 2 workspaces), Hidden1 (7 objects, 1 workspace), Covered1 (7 objects, 1 workspace), and Complex1 (15 objects, 1 workspace). For each scenario, shown in Fig. 6, we conducted 20 trials and reported statistical results, including a 95% confidence interval for discounted cumulative reward, steps, and success rate within 50 steps, as shown in Table 1. For all methods reported in Table 1, the time allocated for each planning step is capped at 60 seconds. Our method consistently outperforms others across all scenarios, thanks to efficient neural filtering and refined clustering. To illustrate the process, we present the action sequence for the Covered1 scenario, completed in 18 steps by adjusting robot configurations, identifying obstacles and target objects, and removing them (highlighted by yellow dashed circles) as shown in Fig. 5. Each step also includes the odds value of the grid world $Odd(\mathcal{G})$ (lower left) and the detected camera image (lower right). The odds values converge near the target object. More details and results are shown in Appendix K.

**Experimental results:** We applied it to both the Stretch robot simulator and its real-world platform (Fig. 7) for validation. Simulation results are in Appendix L. While object-level primitives are assumed to be fully implemented (which is challenging in reality), performance will decrease due to failed actions, and the success rate is not 100%. The goal of the robot is to look for the red bottle. In 18 real-world trials, 10 were successful, and 8 failed due to hardware or communication issues. For successful trials, the discounted cumulative reward is $64890.7 \pm 17760.8$ with $10.6 \pm 3.3$ steps. Visible experimental results are in the video.

Figure 7: The real-world planning environment.

**Limitation**: Our primary limitations stem from errors and failures in perception, execution, and navigation, rather than our focused planning part. First, reliance on pre-existing maps is challenging, as such maps may not be available for real-world robots; integrating advanced SLAM techniques could address this. Second, achieving 100% success for primitive actions in real scenarios is unrealistic, impacting overall performance. Additionally, our point cloud segmentation may produce inaccurate bounding boxes for objects with large contact areas, leading to faulty data association and belief updates. Object detection methods (e.g., YOLO, SIFT, color matching) also struggle in low-light conditions or environments with sparse features. We believe these limitations can be mitigated through advancements in perception, navigation, and execution.

## 6 CONCLUSION

We propose NPF-$k$CT, a novel POMDP framework and solver for 3D object search with hybrid actions. Combining MCTS, NPs, $k$-center clustering, and a revised UCB strategy, it selects optimal actions based on maps, photos, and onboard sensors. Simulations and real-world tests show it outperforms classical solvers, achieving higher rewards, fewer steps, and better success rates within the same computational resources.

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

## A    GRID WORLD

The belief of the position $\boldsymbol{p}_{o^0}^p$ of the guessed target object is saved in a grid world $\mathcal{G}_f$ generated by all workspaces $\mathcal{W}$ using many odds values $Odd(\mathcal{G}_f)$ with a given resolution. Inside each grid, the position is sampled uniformly and its probability is obtained by its odds value. The odds values are updated based on the FOV of the camera using real-world measurement in the excursion process, which is similar to the update of the occupancy grid map Chen et al. (2020); Zhao et al. (2024a), following:

$$\log Odd(g_i|z_{1:t}) = \log Odd(g_i|z_t) + \log Odd(g_i|z_{1:t-1})$$
$$Odd(g_i|z_{1:t}) = P(g_i|z_{1:t})/P(\neg g_i|z_{1:t}) \tag{5}$$

where $P(g_i|z_{1:t})$ and $P(\neg g_i|z_{1:t})$ means the probability of the object belonging to and not belonging to this grid $g_i$ based on multiple observations $z_{1:t}$; $Odd(g_i|z_{1:t})$ is the corresponding odd value. In the belief tree search, we do not update the odds values $Odd(\mathcal{G}_f)$ and it will be updated after real-world excursion and observation using FOV. In the planning stage, this grid world is just used to sample the potential position of the guessed target object in the root node. The guessed target object is special with a constant (no need to estimate) orientation (set as $(0, 0, 0, 1)$), size $(0.1, 0.1, 0.1)$, and move-ability value (set as 100, movable). The grid values $\boldsymbol{g}_{o^0}$ and the declared value $u_{o^0}$ are update-able in the belief tree search but need to be reinitialized as the given value after each excursion. The guessed target object is not the really detected objects. Fig. 8 to show the scenario about grid world for the guessed target object: The other object $\boldsymbol{s}_{o^i}$, $i \neq 0 \in \mathbb{R}^{1 \times 20}$ follows the same format and but

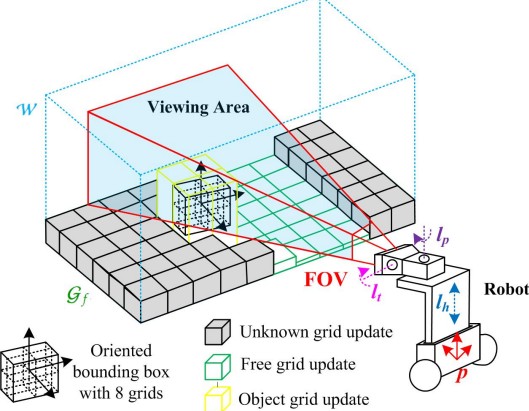

Figure 8: Grid world update in one frame for guessed target object

all parameters should be updated both in belief tree search and real-world excursion. It is noted that the pose of the other objects $\boldsymbol{s}_{o^i}$ is estimated based on the point cloud which is independent of the grid world $\mathcal{G}_f$.

## B    ACTION EXECUTION AND PERCEPTION WITH ON-BOARD SENSORS

Different from many state-of-the-art methods Zheng et al. (2021; 2022) considering the static objects and no interaction between the robot with the target and obstacle objects, our method introduces the robot arm action to remove obstacle objects and free the undetected space. Meanwhile, our perception part is carefully explored with many useful outputs, like estimated object pose, estimated object size, object move-ability, and object detector, fusing both point cloud data and the image data without using manual marks.

### B.1    ACTION EXECUTION

Our framework for object search is suitable for all mobile robots with 2D Lidar and RGBD cameras, but specifically, we mainly consider the Fetch robot here. The move base action is implemented using a ROS interface move_base and the interaction with the AMCL-based navigation stack. The robot lift height and head joints including pan and tilt angles are controlled by following the joint-space trajectories on a group of joints based on a ROS interface ros_control Chitta et al. (2017). The removing action is to pick up the object and place it in some given areas outside the workspace.

Some examples about the move base action, joints controller, and removing object action are shown in Fig. 9.

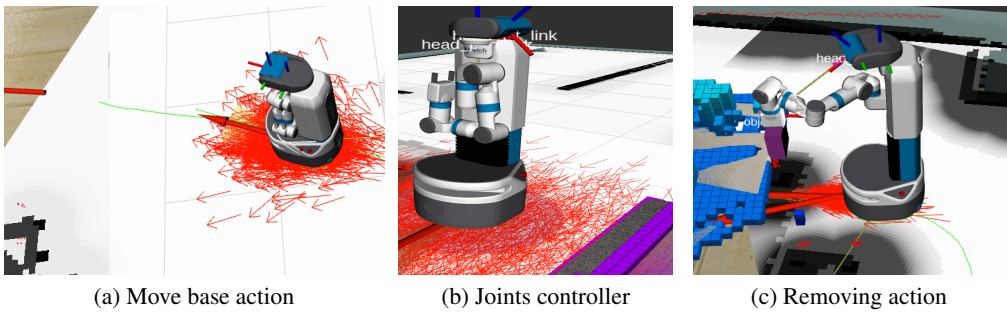

(a) Move base action     (b) Joints controller     (c) Removing action

Figure 9: Actions execution

## B.2 SENSOR DATA OPERATIONS

In our framework, the point cloud map $\mathcal{M}$ of the robot environment with some furniture and known objects, but not all objects, is assumed to be available before planning, which is a reasonable and realistic assumption achieved by mapping the environment at any time before the tasks. In the online planning and execution process, the point cloud $\mathcal{F}_j$ in $j$-th frame detected from the depth camera is fused with the map $\mathcal{M}$ by iterative closest point (ICP) with random sample consensus (RANSAC). The computed ICP transformation also helps to give a noisy measurement $\boldsymbol{Z}_{icp} \sim \mathcal{N}(\bar{\boldsymbol{Z}}_{icp}, \sigma_{icp})$ for robot pose the in the global frame, which will be fuse with the AMCL localization $\boldsymbol{Z}_{amcl} \sim \mathcal{N}(\bar{\boldsymbol{Z}}_{amcl}, \sigma_{amcl})$ in the filter part of the POMDP framework. With the increase in the frame number, the point cloud of objects and environment $\mathcal{P}_j = \mathcal{M} \bigcup \mathcal{F}_0 \bigcup \cdots \bigcup \mathcal{F}_j$ is becoming more and more complete. After removing the original map $\mathcal{M}'_c = \mathcal{P}_j / \mathcal{M}$, the point clouds for $n$ newly detected objects $\{o_0,\ o_1, \cdots,\ o_n\} \in \mathcal{M}'_c$ are extracted by point cloud segmentation using the Euclidean cluster extraction method. Then, in order to estimate the object pose and size for performing further manipulator interaction, the minimum oriented bounding box for each object is obtained by principal component analysis. The above point cloud segmentation is implemented both on local and global point clouds and then a data association, based on the Mahalanobis distance of their centriod points and the point-wise mean distance, is introduced. The process to get measurement from point clouds is shown in Fig. 10.

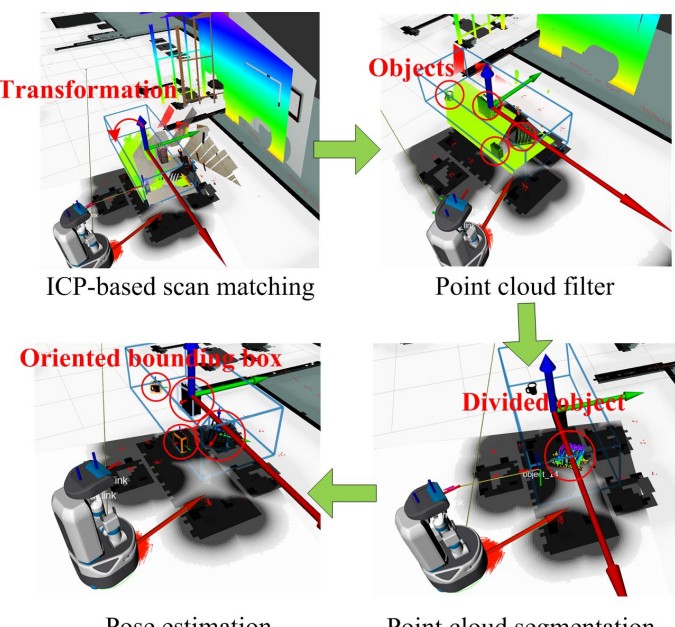

ICP-based scan matching      Point cloud filter

Pose estimation      Point cloud segmentation

Figure 10: Measurement from point cloud

### B.3 OBJECT DETECTOR

State-of-the-art real-time object detection systems, like YOLO, are commonly designed to divide the objects into different classes and they are not matched with the target images. Meanwhile, we have the 3D point clouds of the objects, which are helpful to divide the objects in the image. So as to complete the given object detection task using several given images and some semantic words (optional), we fuse the traditional feature-matching method and YOLO toolbox to complete the object detection task.

Based on the previous point cloud segmentation, we perform it on the current visual local frame and the separated point clouds in the local frame are re-projected to the image to bound the objects in the RGB camera image forming a set of sub-images $\{\mathcal{I}_i^p,\ i = 1,\ 2, \cdots,\ n\}$ using the camera configuration and the perspective projection. Similar sub-images of this image $\{\mathcal{I}_i^y,\ i = 1,\ 2, \cdots,\ m\}$ and their corresponding semantic scores $\{\bar{s}_i^y,\ i = 1,\ 2, \cdots,\ m\}$ can also be bounded and generated using the YOLOv5 model with pre-trained parameters. Commonly, we have $m \neq n$. A simple data association method with the nearest images and enough common areas is presented to match these two sets of sub-images. For the successful data association pair, we use the sub-image in the local frame as the image corresponding to this object. These sub-images in the detected and associated 2D boxes corresponding to different objects are matched with the target object using SIFT descriptor. The rate between the number of matched scale-invariant features and the number of all features is defined as the probability of object detection, denoted $\{\bar{s}_i^d,\ i = 1,\ 2, \cdots,\ n\}$. If this task offers the target type, like cup and laptop, we use the mean values between the semantic scores $\{\bar{s}_i^y,\ i = 1,\ 2, \cdots,\ n\}$ and the probability of object detection $\{\bar{s}_i^d,\ i = 1,\ 2, \cdots,\ n\}$. The main process of the object detector is given in Fig. 11. We find the current object matching is not robust enough in the real world environment and the perception is not our main focused point, so we also add the color matching for the detected object in real world experiments [7] when the offered object detector fails.

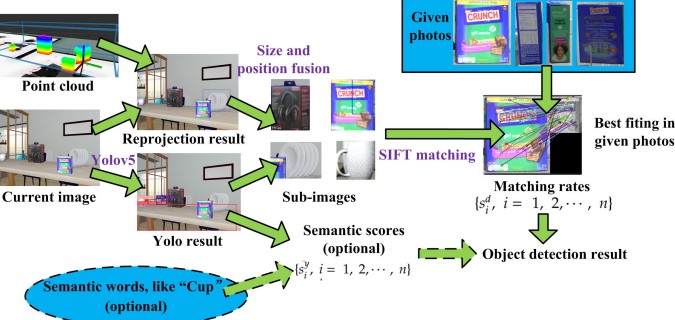

Figure 11: Object detector

### B.4 MOVE-ABILITY ESTIMATION

It is easy to know that, in the real-world environment, some objects in the workspace are not moveable for the robot with a manipulator due to some physical limitations, such as the size limitation of the object, the manipulator workspace limitation, and the mobile base motion limitation. In our framework, we would like to manipulate the objects in the workspace to free some FOV, so it is better to estimate the probability of the move-ability and then update their beliefs for POMDP planning.

Based on the point cloud segmentation for the fused global point cloud, we can obtain many separated point clouds for different objects. Then, facing each point cloud in the detected frame, many candidate grasp poses are predicted by the learning-based Grasp Pose Detection (GPD) toolbox Ten Pas et al. (2017). So as to reduce the computational complexity, we select $k$ representative grasp poses $\boldsymbol{p}_i^g,\ i = 1, 2, \cdots, k$ for each object using k-means clustering algorithm. These $k$ grasp poses are diverse with high scores in picking success rate. The point clouds of the obstacles in the surrounding environment and these $k$ representative grasp poses $\boldsymbol{p}_i^g$ are transformed to the local frames $T_r^g(\boldsymbol{p}_i^g)$ based on the pre-visited robot poses $\boldsymbol{p}_i^r$ during the task process. Here, it is noted

---

[7]In real-world experiments, we commonly use the target object with a large area of pure color, like the pure red bottle.

that only the pre-visited robot poses are considered because the poses generated by other methods may not reachable based on the used move-base toolbox because of the error of the AMCL localization and the complexity of the occupancy grid map. These pre-visited robot poses $\boldsymbol{p}_i^r$ are safer for implementation. Following, these transformed local poses $T_r^g(\boldsymbol{p}_i^g)$ will be set as the plan target to the robot manipulator using moveit toolbox without execution in a given time limitation $t_m$. The planned moveit feedback will decide the probability of this detection about the move-ability $0 < r_{move} \leq 1$ based on distance. Otherwise, it will be set as 1. When no solution for moveit toolbox, the move-ability $r_{move}$ will get close to 0. In the planning stage, for each particle, we will randomly sample a random value for this object and compare it with the move-ability $r_{move}$ to identify the move-ability in this step. Objects with too large sizes will be considered to be non-moveable $r_{move} = 0$, which is definitely not movable. In real-world experiments, for simplification, we use all removable objects and the move-ability $r_{move}$ is set to be 1. An example of the candidate grasp poses is shown in Fig. 12.

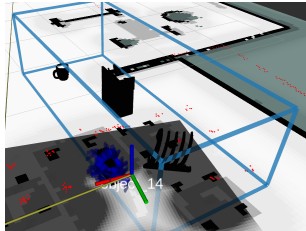

Figure 12: Candidate grasp poses

## C    PROOF ABOUT THEOREM 1

*Proof.* We rewrite the formulation by $g'(\theta) \triangleq g(\boldsymbol{a}, \boldsymbol{s}, \alpha(\mathcal{P}))$ given fixed $\boldsymbol{s}$ and $\alpha(\mathcal{P})$. Based on $g'(\theta) \sim nn(\mu(\theta), \sigma(\theta))$, with the fix input $\theta_i$ and $i \leq 1$, we have the Gaussian distribution $g'(\theta_i) \sim N(\mu(\theta_i), \sigma(\theta_i))$.

Let $z_i = \frac{g'(\theta_i) - \mu(\theta_i)}{\sigma(\theta_i)} \sim N(0,1)$. For a Gaussian distribution with mean 0 and variance 1, we have:

$$
\begin{aligned}
Pr(z_i > \eta_i) &= \int_{\eta_i}^{+\infty} \frac{1}{\sqrt{2\pi}} \exp^{-z^2/2} dz \\
&= \int_{\eta_i}^{+\infty} \frac{1}{\sqrt{2\pi}} \exp^{-(z-\eta_i)^2/2 - z\eta_i + \eta_i^2/2} dz \\
&= \exp^{-\eta_i^2/2} \int_{\eta_i}^{+\infty} \frac{1}{\sqrt{2\pi}} \exp^{-(z-\eta_i)^2/2 - z\eta_i + \eta_i^2} dz \\
&= \exp^{-\eta_i^2/2} \int_{\eta_i}^{+\infty} \frac{1}{\sqrt{2\pi}} \exp^{-(z-\eta_i)^2/2} \exp^{-z\eta_i + \eta_i^2} dz
\end{aligned}
\tag{6}
$$

Because we can set $\eta_i > 0$ and $z_i > \eta_i$, we have: $-z\eta_i + \eta_i^2 < 0$. So, we have:

$$
Pr(z_i > \eta_i) \leq \exp^{-\eta_i^2/2} \int_{\eta_i}^{+\infty} \frac{1}{\sqrt{2\pi}} \exp^{-(z-\eta_i)^2/2} dz = \exp^{-\eta_i^2/2}/2
\tag{7}
$$

Introduce $z_i = \frac{g'(\theta_i) - \mu(\theta_i)}{\sigma(\theta_i)}$, we have:

$$
\begin{aligned}
Pr(g'(\theta_i) - \mu(\theta_i) > \eta_i \sigma(\theta_i)) &\leq \exp^{-\eta_i^2/2}/2 \\
Pr(g'(\theta_i) - \mu(\theta_i) < -\eta_i \sigma(\theta_i)) &\leq \exp^{-\eta_i^2/2}/2
\end{aligned}
\tag{8}
$$

We have:

$$
Pr(\|g'(\theta_i) - \mu(\theta_i)\| > \eta_i \sigma(\theta_i)) \leq \exp^{-\eta_i^2/2}
\tag{9}
$$

The complementary set of $\|g'(\theta_i) - \mu(\theta_i)\| > \eta_i \sigma(\theta_i)$ is $\|g'(\theta_i) - \mu(\theta_i)\| \leq \eta_i \sigma(\theta_i)$ and its corresponding probability is bigger than $1 - \exp^{-\eta_i^2/2}$. Let $\delta = \exp^{-\eta_i^2/2} \Rightarrow \eta_i = \sqrt{-2\log(\delta)}$, we have:

$$Pr(\|g'(\theta_i) - \mu(\theta_i)\| \leq \eta_i \sigma(\theta_i)) \geq 1 - \delta \tag{10}$$

We have the scoring function $g'(\theta_i)$ is bounded by $(\mu(\theta_i) - \sqrt{-2\log(\delta)}\sigma(\theta_i), \mu(\theta_i) + \sqrt{-2\log(\delta)}\sigma(\theta_i))$ with a probability bigger than $1 - \delta$. So, if its lower bound $\mu(\theta_i) - \sqrt{-2\log(\delta)}\sigma(\theta_i) > 0 \Leftrightarrow \mu(\theta_i) > \sqrt{-2\log(\delta)}\sigma(\theta_i)$, the scoring function satisfying:

$$Pr(g'(\theta_i) > 0) > 1 - \delta \tag{11}$$

$\square$

## D STRUCTURAL GRAPH OF NPF-$k$CT

NPF-$k$CT follows a standard procedure with four alternating stages: planning, execution, obtaining observations, and filtering, as shown in Algorithm 1. Its flow chart is shown as follows:

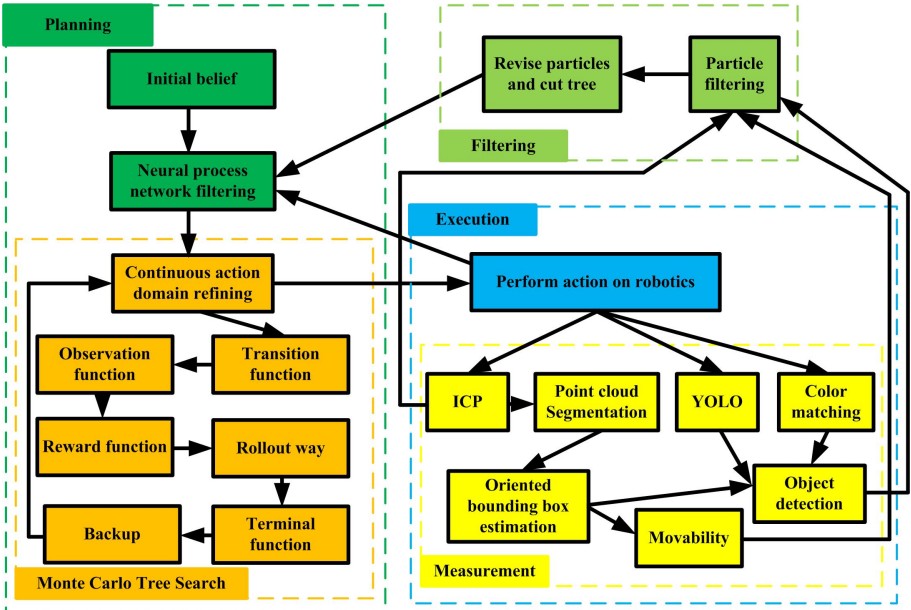

Figure 13: The main steps in NPF-$k$CT

## E ALGORITHM TO GET THE SIMULATION DATASET

Algorithm 4 shows how to use Gazebo simulator to collect the data for scoring function. The input of method including the 2D occupancy grid map, which is used to initialize the 3D ICP matching, 3D point cloud map, which is used to get fused object point cloud by completing the scan matching with current 3D camera point cloud and remove the points outside the workspace, Fetch simulator, which offers all real-time sensor data, and some manually selected candidate actions $\mathcal{A}_{manual}$. These selected candidate actions $\mathcal{A}_{manual}$ are selected from some random generated actions with good diversity. The outputs are the pairs for neural network mapping from robot state $s_r$, generated grid world $\mathcal{G}_f$, the detected objects in workspace with a point cloud format $\mathcal{M}_c'$, and the 8 grids status of the target object compared with thresholds $g_{o^i}$ to the scoring value, which is the probability of updating the grids of the target object. The whole process is shown in Algorithm 4.

## F NETWORK STRUCTURE FOR SCORING FUNCTION

The structure of the used NPs model for the scoring function is shown in Fig. 15.

---

**Algorithm 4** Simulation dataset generator

---

**Input:** 2D occupancy grid map, corresponding 3D point cloud map, Fetch robot simulator in Gazebo environment, a set of manually selected candidate actions $\mathcal{A}_{manual}$

**Output:** data mappings $(\boldsymbol{s}_r, \, \mathcal{G}_f, \, \mathcal{M}'_c, \, \boldsymbol{g}_{o^i}) \rightarrow g(\boldsymbol{a}, \, \boldsymbol{s}, \, \alpha(\mathcal{P}))$

1: **while** Dataset size is not enough **do**
2:    Re-initialize the robot simulator and Gazebo environment with different object numbers and poses.
3:    Randomly generate $J$ classes of action sequences $\{\boldsymbol{a}_1^j, \, \boldsymbol{a}_2^j, \ldots, \, \boldsymbol{a}_L^j\}$, $\boldsymbol{a}_i^j \in \mathcal{A}_{manual} \subseteq \mathcal{A}$, $j = 1, \, 2, \cdots, N$ from $\mathcal{A}_{manual}$ with limited $I$ steps.
4:    **for** $i = 1$ to $I$ **do**
5:        **for** $j = 1$ to $J$ **do**
6:            Set robot status based on the action $\boldsymbol{a}_i^j$ with some noises.
7:            Collect camera point cloud $\mathcal{M}_i^j$ and fuse point cloud $\mathcal{M}'_c = f_F(\mathcal{M}'_c \bigcup \mathcal{M}_i^j)$ after ICP and filtering the point cloud outside the workspace.
8:            Compute the odds update for the whole grid world about the grid world for guest target object $\mathcal{G}_f$ based on FOV and object detection.
9:            Remove the target object (identified) and undetected objects in Gazebo environment to make sure that we just use the known information for data generation.
10:       Collect and save $\mathcal{M}'_c$ and $Odd(\mathcal{G}_f)$.
11:       **for** $k = 1$ to $K$ **do**
12:           Uniformly sample configurations in continuous action domain $\boldsymbol{s}_r \sim \text{Uniform}(\mathcal{A}_c)$.
13:           Build a cube with 8 both red and green grids. The color distribution is decided by the comparison value $f_c(\boldsymbol{g}_{o^0}) \in \{0, \, 1\}^{1 \times 8}$ between $\boldsymbol{g}_{o^0}$ and threshold.
14:           Collect and save $\boldsymbol{s}_r$ and $f_c(\boldsymbol{g}_{o^0})$.
15:           $score_d \leftarrow 0$
16:           **for** $l = 1$ to $L$ **do**
17:               Sample the position of the cube with 8 grids in different positions based on odds value $Odd(\mathcal{G}_f)$ and grid colors are set based on $f_c(\boldsymbol{g}_{o^i})$. Only the unobserved grids, of which the value is smaller than threshold, are set as green for matching. Otherwise, the grids are set as red.
18:               **if** Object detection finds the green area based on the collected RGBD image is True **then**
19:                   $score_d \leftarrow score_d + 1$
20:               **end if**
21:           **end for**
22:           $g(\boldsymbol{a}, \, \boldsymbol{s}, \, \alpha(\mathcal{P})) \leftarrow \frac{score_d}{L} \times 100\%$.
23:           Collect and save $g(\boldsymbol{a}, \, \boldsymbol{s}, \, \alpha(\mathcal{P}))$.
24:       **end for**
25:       Rearrange all objects based on their original poses before removing them.
26:       **end for**
27:    **end for**
28:    **for** $m = 1$ to $M = I \times J \times K \times L$ **do**
29:        Normalize the following data mappings: $(\boldsymbol{s}_r, Odd(\mathcal{G}_f), \mathcal{M}'_c, \boldsymbol{g}_{o^0}) \rightarrow g(\boldsymbol{a}, \, \boldsymbol{s}, \, \alpha(\mathcal{P}))$.
30:    **end for**
31:    **return** Saved data pairs by tensor format
32: **end while**

---

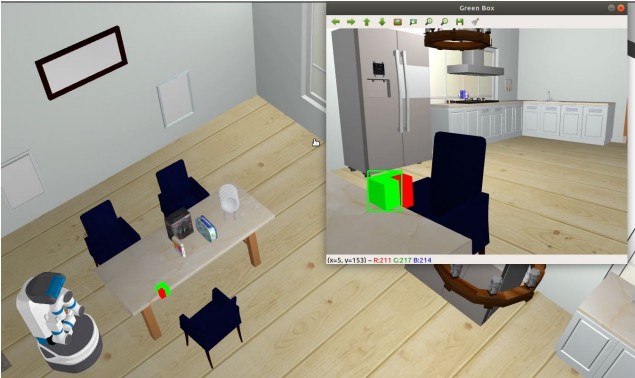

Figure 14: Robot successfully detects the green grids in this scene and "$score_d$" adds 1 in Algrithm 4.

## G NETWORK TO SAMPLE THE CANIDATE ACTIONS

The main algorithm to get the samples $\{a_i\}$ is shown in Algorithm 5.

---

**Algorithm 5** Net_sam($Odd(\mathcal{G}_f)$, $\mathcal{P}_j$, $\boldsymbol{b}$,)

---

**Input:** The trained network $nn(\mu(\boldsymbol{a}), \sigma(\boldsymbol{a}))$, the odds update for the whole grid world about the fake object $Odd(\mathcal{G}_f)$, belief $\boldsymbol{b} = \{s_i\}$, fused point cloud $\mathcal{M}'_c$, throushold $C_{threshold}$
**Output:** A set of potential actions $\{a_i\}$ satisfy the following condition $Pr[g(\boldsymbol{a}_i, \boldsymbol{s}_i, \alpha_i(\mathcal{P})) > 0, \forall i] \geq C_{threshold}$
 1: Transform odds value for the grid world as an image and save as repeated tensor $T_{odd}$.
 2: Repeats the fused point cloud and saves it as a tensor $T_{point}$.
 3: Samples a class of states $\{s_i\}$.
 4: $\{s_r\}$, $\{g_{o^0}\} \leftarrow \{s_i\}$ and gets the robot state tensor $T_r$ and 8 grids odds tensor $T_g$.
 5: Gets the predicted mean $T_\mu$ and variance tensors $T_\sigma$ based on $T_r, T_{odd}, T_{point}, T_g$ and the trained network $nn(\mu(\boldsymbol{a}), \sigma(\boldsymbol{a}))$.
 6: **for** $\mu(\boldsymbol{a}_i) \leftarrow T_\mu, \sigma(\boldsymbol{a}_i) \leftarrow T_\sigma$ **do**
 7: $\quad \beta_i^* \leftarrow (2\log(1/(1 - C_{threshold}))^{\frac{1}{2}}$
 8: $\quad$ Check $\mu(\boldsymbol{a}) > \beta^*\sigma(\boldsymbol{a})$ and collect the ones satisfying this condition to $\{a_i\}$.
 9: $\quad$ **if** satisfying the number limitation **then**
10: $\quad\quad$ **return** selected action set $\{a_i\}$.
11: $\quad$ **end if**
12: **end for**
13: **return** action set $\{a_i\}$ with highest score.

---

## H BACKUP

When each episode reaches the terminal state, our NPF-$k$CT framework updates the estimation reward $\hat{Q}(\boldsymbol{node}_o, \boldsymbol{a})$ as well as the visited numbers $N(\boldsymbol{node}_o)$ and $N(\boldsymbol{node}_o, \boldsymbol{a})$ of all nodes visited by this episode. Here, we present two classical stochastic backup methods including the Bellman backup (Algorithm 6), which is used in the ABT method and similar to the rule used in Q-learning, and the Monte-Carlo backup (Algorithm 6), which is widely used in many outstanding POMDP solvers, like POMCP, POMCPOW, and VOMCPOW. The Bellman update naturally follows the objective function of the POMDP formulation that aims to pursue optimal action in each step of the long-term planning. It helps the solver to explore deeper by focusing its search on promising parts of the belief tree. The main challenge for the Bellman backup is when facing unexpected observations, a lot of deeply explored belief trees will be frequently cut and this case causes poor planning performance. Hence, the Bellman backup gets a better performance when the good rewards are sparse in the belief tree, but it is not stable enough for the poor observation prediction. In contrast to selecting the reward with optimal action, the Monte-Carlo backup computes the average reward along with different action episodes, which means that the generated belief tree will be more robust when facing unexpected uncertainty in received observation. In our application for object search, the real visual observation is not well predictable for the observation model in the POMDP formulation and the robot camera will frequently receive unexpected measurements, which may not be deeply

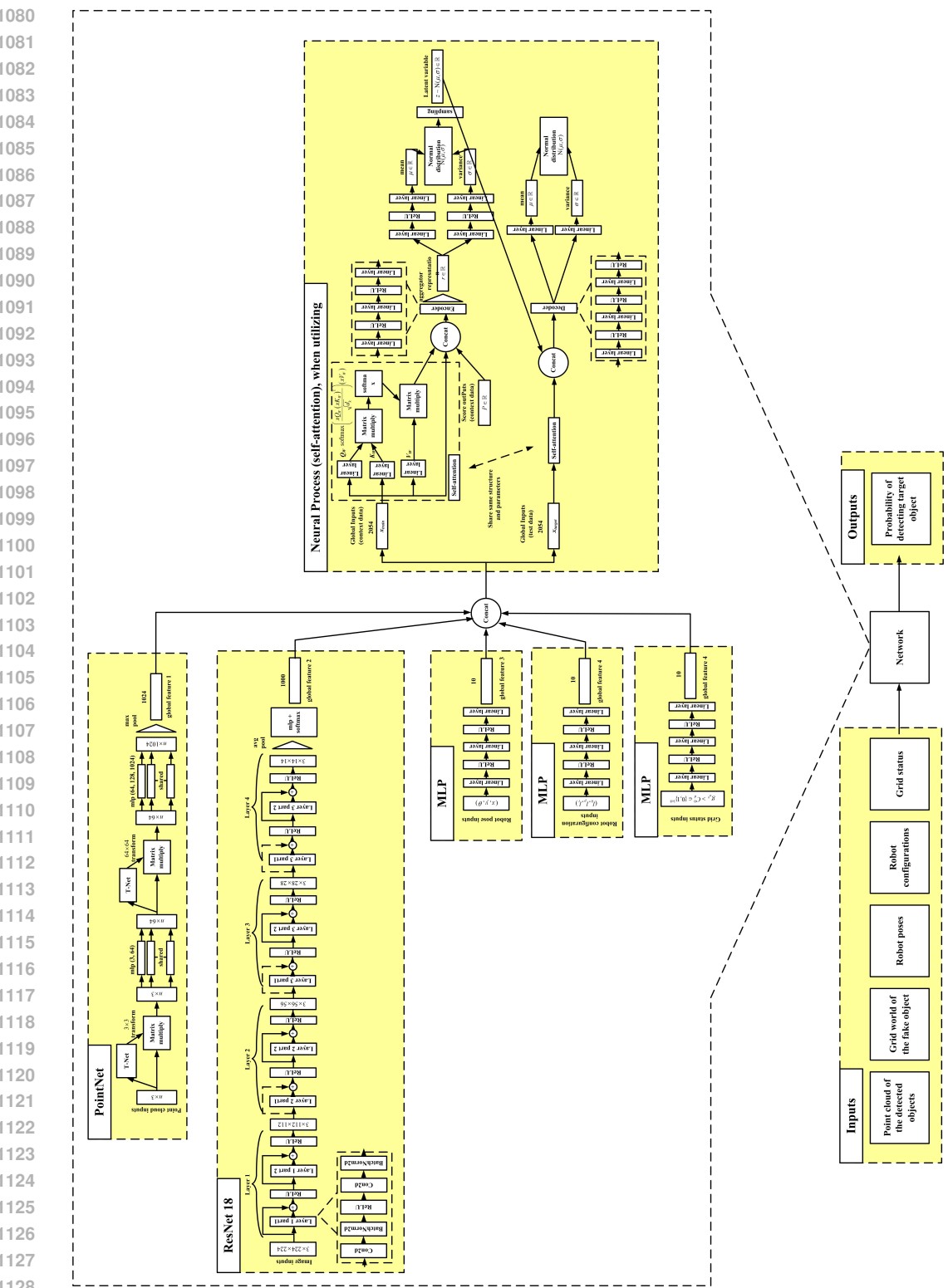

Figure 15: Used network.

explored in the belief tree and breaks the advantage of the Bellman update. We use the Monte-Carlo backup in our problem but the Bellman backup is commonly superior in other applications, so we present both two backup ways here.

---

**Algorithm 6** Backup($\mathcal{T}$, $\boldsymbol{node}_o$, $\boldsymbol{a}^*$, $r$, $R$)

---

**Input:** The belief tree $\mathcal{T}$, the observation mode $\boldsymbol{node}_o$, the selected action $\boldsymbol{a}^*$, the accumulated reward $r$, and the immediate reward $R$
**Output:** The updated tree $\mathcal{T}$ with updated values
1: $N(\boldsymbol{node}_o) \leftarrow N(\boldsymbol{node}_o) + 1$ and $N(\boldsymbol{node}_o, \boldsymbol{a}^*) \leftarrow N(\boldsymbol{node}_o, \boldsymbol{a}^*) + 1$
2: **if** Use Monte-Carlo backup **then**
3:     $\hat{Q}(\boldsymbol{node}_o, \boldsymbol{a}^*) \leftarrow \hat{Q}(\boldsymbol{node}_o, \boldsymbol{a}^*) + \frac{r - \hat{Q}(\boldsymbol{node}_o, \boldsymbol{a}^*)}{N(\boldsymbol{node}_o, \boldsymbol{a}^*)}$
4: **else**
5:     $\boldsymbol{node}'_o$ is the child of $\boldsymbol{node}_o$
6:     $\hat{Q}(\boldsymbol{node}_o, \boldsymbol{a}^*) \leftarrow \hat{Q}(\boldsymbol{node}_o, \boldsymbol{a}^*) + \frac{R + \gamma \hat{V}^*(\boldsymbol{node}'_o) - \hat{Q}(\boldsymbol{node}_o, \boldsymbol{a}^*)}{N(\boldsymbol{node}_o, \boldsymbol{a}^*)}$
7:     $\hat{V}^*(\boldsymbol{node}_o) \leftarrow \max_{\boldsymbol{a} \in \mathcal{L}(\boldsymbol{node}_o)} \hat{Q}(\boldsymbol{node}_o, \boldsymbol{a})$
8: **end if**

---

# I  THEORETICAL ANALYSIS

In this section, we aim to analyze the convergence of the proposed solver with some assumptions to improve the compact of this work. The key point is to answer the following question:

**Question 1.** *Does the NPF-$k$CT algorithm converge in probability to the optimal value function in POMDPs?*

Let's recall our key steps in NPF-$k$CT algorithm related to the convergence, including network filtering, $k$ clustering for hierarchical partition, and the revised UCB strategy. It is easy to know that the prediction accuracy of the neural network will directly affect the performance of the method.

**Assumption 1.** *The neural network used for action filtering does not filter the optimal action. The obtained feasible region $\mathcal{X}$ satisfies $\boldsymbol{a}^* \in \mathcal{X}$, $\boldsymbol{a}^* = \boldsymbol{argmax}_{\boldsymbol{a} \in \mathcal{A}} Q(\boldsymbol{b}, \boldsymbol{a})$.*

Satisfying assumption 1, the network filtering will not affect the convergence of the NPF-$k$CT algorithm to the optimal result. The problem just has a smaller action domain with the same configuration. To answer Question 1 is equal to answer Question 2:

**Question 2.** *Without considering network filtering, does the NPF-$k$CT algorithm with $k$-center clustering and the revised UCB strategy converge in probability to the optimal value function in POMDPs?*

Based on Silver & Veness (2010), we can answer Qusetion 2 by considering POMDPs as a derived MDP. Let's consider Lemma 1 about the value function and Lemma 2 about the rollout distribution Silver & Veness (2010):

**Lemma 1.** *Given a POMDP $\mathcal{M} = <\mathcal{S}, \mathcal{A}, \mathcal{O}, T, Z, R>$[8] consider the derived MDP with histories as states, $\widetilde{\mathcal{M}} = <\mathcal{H}, \mathcal{A}, \widetilde{T}, \widetilde{R}>$, where $\widetilde{T}(h, \boldsymbol{a}, hao) = \sum_{\boldsymbol{s} \in \mathcal{S}} \sum_{\boldsymbol{s}' \in \mathcal{S}} \boldsymbol{b}(\boldsymbol{s}, h) T(\boldsymbol{s}, \boldsymbol{a}, \boldsymbol{s}') Z(\boldsymbol{s}', \boldsymbol{a}, \boldsymbol{o})$, where $\boldsymbol{b}(\boldsymbol{s}, h) = Pr(\boldsymbol{s}|h)$, $h$ is the given history, hao means the updated history pruning the tree by $\boldsymbol{a}$ and $\boldsymbol{o}$, and $\widetilde{R}(h, \boldsymbol{a}) = \sum_{\boldsymbol{s} \in \mathcal{S}} \boldsymbol{b}(\boldsymbol{s}, h) R(\boldsymbol{s}, \boldsymbol{a}, \boldsymbol{s}') = \sum_{\boldsymbol{s} \in \mathcal{S}} \boldsymbol{b}(\boldsymbol{s}, h) R(\boldsymbol{s}, \boldsymbol{a})$. Then the value function $\bar{Q}^\pi(h)$ of the derived MDP is equal to the value function $Q^\pi(h)$ of the POMDP, $\forall$ policy $\pi$ $\bar{Q}^\pi(h) = Q^\pi(h)$.*

**Lemma 2.** *For any rollout policy $\pi$, the POMDP rollout distribution is equal to the derived MDP rollout distribution, $\forall \pi$ $D^\pi(h_T) = \widetilde{D}^\pi(h_T)$.*

Based on Lemma 1 and Lemma 2, we can find that proving the convergence of POMDP solvers for a given POMDP is equal to proving the convergence of corresponding MDP solvers for the driver MDP. Hence, we have the following new question to replace Question 2:

**Question 3.** *Does the NPF-$k$CT algorithm converge in probability to the optimal value function in MDPs?*

In order to connect our NPF-$k$CT algorithm with some existing MDP solvers, we present the following assumption:

---
[8]ignore initial belief $\boldsymbol{b}_0$ and the discounted factor $\gamma$ here.

**Assumption 2.** *The action partitioning results $\mathcal{X}_{d,i_j}$, $j = 1, \cdots, k$ generated by all $k$ center clustering operations for the action domain $\mathcal{X}_{d-1,i}$ follow the properties of the hierarchical partitioning, satisfying $\mathcal{X}_{d,i_j} \bigcap \mathcal{X}_{d,i'_j} = \emptyset$ for $\forall i_j, i'_j \in \{i_1, \cdots, i_k\}$ and $\bigcup_{j=1,\cdots,k} \mathcal{X}_{d,i_j} = \mathcal{X}_{d-1,i}$.*

Based on Assumption 2, we can consider the $k$ center clustering as the hierarchical partitioning. With hierarchical partitioning, our method follows the same search strategy as the HOO method but with different partitioning ways. Because we limit the refining accuracy $range_i$ and the corresponding list dimension $|\mathcal{L}(node_o)|$, we can consider the same problem with the finite discrete actions and each action $\boldsymbol{a}$ is a range instead of a value. We have the following new questions:

**Question 4.** *Does the action selection strategy equation 3 in the NPF-kCT algorithm converge in probability to the optimal value function in MDPs with discrete action domains?*

When the number of visits $N(\boldsymbol{node}_o)$ approaches infinity, the action range $range_i$ will be the constant limitation $D_{lim} \in \mathbb{R}$. For the action selection strategy equation 3, with a given coefficient $\omega_2$, we can ignore the region-related terms $\omega_2 range_i$ due to the same constant value for all candidate action ranges. In this way, the action selection strategy becomes the standard UCB1 bound:

$\hat{Q}(\boldsymbol{node}_o, \boldsymbol{a}) + \omega_1 \sqrt{\frac{\log N(\boldsymbol{node}_o)}{N(\boldsymbol{node}_o, \boldsymbol{a})}} + \omega_2 range_i \to \hat{Q}(\boldsymbol{node}_o, \boldsymbol{a}) + \omega_1 \sqrt{\frac{\log N(\boldsymbol{node}_o)}{N(\boldsymbol{node}_o, \boldsymbol{a})}} + \omega_2 D_{lim}$.

So it follows the convergence analysis for the UCB1 in Kocsis & Szepesvári (2006) and Silver & Veness (2010), following:

**Lemma 3.** *For a suitable choice of $\omega_1$, the value function constructed by UCT converges in probability to the optimal value function. As the number of visits $N(node_o)$ approaches infinity, the bias of the value function is $O(\log N(node_o)/N(node_o))$.*

This convergence result means that the method can find the optimal range action $\mathcal{A}_{opt}$ that has the largest mean value for all refined ranges with some probability.

**Assumption 3.** *The obtained mean values $\bar{Q}(\boldsymbol{b}, \mathcal{A}_{opt})$ and $\bar{Q}(\boldsymbol{b}, \mathcal{A}_{sub})$ corresponding the optimal action range $\mathcal{A}_{opt}$ and any sub-optimal action range $\mathcal{A}_{sub}$ satisfy:*

$$\bar{Q}(\boldsymbol{b}, \mathcal{A}_{opt}) - \bar{Q}(\boldsymbol{b}, \mathcal{A}_{sub}) \geq \eta D_{lim} \tag{12}$$

Based on Lipschitz continuous, we have any action $\boldsymbol{a} \in \mathcal{A}_{sub}$ in the sub-optimal range $\mathcal{A}_{sub}$ satisfies: $Q(\boldsymbol{b}, \boldsymbol{a}) \leq \bar{Q}(\boldsymbol{b}, \mathcal{A}_{sub}) + \eta D_{lim}$. Then, considering Assumption 3, we will have: $\bar{Q}(\boldsymbol{b}, \mathcal{A}_{opt}) \geq \eta D_{lim} + \bar{Q}(\boldsymbol{b}, \mathcal{A}_{sub}) \geq Q(\boldsymbol{b}, \boldsymbol{a})$. Because the best action $\widetilde{\boldsymbol{a}}^* \in \mathcal{A}_{opt}$ in the optimal action range $\mathcal{A}_{opt}$ satisfy $Q(\boldsymbol{b}, \widetilde{\boldsymbol{a}}^*) \geq \bar{Q}(\boldsymbol{b}, \mathcal{A}_{opt})$, finally, for any action $\boldsymbol{a} \in \mathcal{A}_{sub} \bigcup \mathcal{A}_{opt}$ in both sub-optimal range $\mathcal{A}_{sub}$ and optimal range $\mathcal{A}_{opt}$, we have: $\widetilde{\boldsymbol{a}}^* = \boldsymbol{a}^*$ and $Q(\boldsymbol{b}, \widetilde{\boldsymbol{a}}^*) = Q(\boldsymbol{b}, \boldsymbol{a}^*) \geq Q(\boldsymbol{b}, \boldsymbol{a})$, which means the obtained optimal range $\mathcal{A}_{opt}$ definitely includes the optimal action $\boldsymbol{a}^*$. In short, based on previous assumptions, the NPF-kCT method can converge in probability to the small range including the optimal solution for POMDPs with continuous action domains.

## J  CONFIGURATIONS FOR SIMULATIONS AND EXPERIMENTS

In Section 5, we utilize the Fetch robot simulator to validate the effectiveness of our approach within the Gazebo environment based on C++ and Python codes. The neural process network undergoes training for a total of 3000 iterations (about 4 hours), executed on a single NVIDIA 3090 GPU. Post-training, we execute our project on a desktop machine, utilizing only the CPU, operating on Ubuntu 18.04, and powered by an Intel Core i7-13700k processor. The evaluation of our methodology spans diverse object configurations in various scenarios, with a comparative comparison against several technologies, including continuous action domain benchmark methods (POMCPOW and Voronoi Optimistic Monte Carlo Planning with Observation Weighting, VOMCPOW) and classical POMDP methods with manual-setting discrete action domain (POMCP and GPOMCP)[9].

---

[9]For the POMCP and GPOMCP methods, the changing robot configuration is divided into 4 manually selected robot poses, 3 candidate robot lift heights, 9 candidate robot head orientations, which can not cover the whole continuous action domain $\mathcal{A}$. Therefore, the comparison between discrete and continuous action methods is not very fair due to the manual selection of the discrete actions and the separation of the base and head actions, but we can directly see the benefit of a wider continuous action domain from these experiments.

The simulations are conducted within a realistic Gazebo living environment, featuring diverse object configurations. Navigation is facilitated by a point cloud map and a 2D occupancy grid map, incorporating furniture and certain known objects, though not encompassing all objects present. The surfaces of the large furniture form the workspaces and some unknown objects including a target object and some obstacles are set inside the workspaces with different positions. All methods are tested in 5 different scenarios. Considering the blue snack box as the target object, the candidate position spaces for the robot are 4 red rectangles surrounding each workspace and the orientation is unrestricted. Expect for the parameter sensitive analysis, the cluster number $k$ for all simulations and experiments is set as 3 [10]. The candidate robot head motion is confined to $l_p \in (-\frac{\pi}{12}, \frac{\pi}{12})$, $l_t \in (-0.5, 0.5)$, and the lift motion $l_h$ is constrained within $(0.0, 0.4)$. In the context of POMDP models for exploration, the hexahedron FOV is defined by a 60-degree horizontal view angle and a vertical range with a height-width ratio of 480/600. The nearest and farthest planes to the camera center are set at 0.5 meters and 1.7 meters, respectively. The grid size of the grid world for the guessed target object is set as 2 cm. Key parameters include grid updating thresholds $v_p = 0.1$ and $v_n = -0.1$, re-initialized grid values for the guessed target object set at 0.2, and reward values following $R_{max} = 10^5$, $R_{c_t} = 5 \times 10^4$, $R_{c_o} = 10^4$, $R_{min} = -1$, and $R_{ill} = -10^3$.

Our solver relies on 7 parameters, including the refining clustering number $k$, coefficients $\omega_1$ and $\omega_2$ for the MCTS action selection strategy, a self-defined exploration constant $C_r$, the minimum radius for partitioning $D_{lim}$, and two coefficients controlling the refining velocity, with $0 \le \bar{\omega}_2 < \bar{\omega}_1 \le 1$. Most of these parameters were not fine-tuned for optimal performance; instead, they were quickly identified or chosen intuitively. Here, we would like to add some explanations to help the users to quickly determine the parameters within a short time, like 20 minutes. Let's go through each parameter:

**Refining Clustering Number $k$:**

Fig. 18 shows that $k = 3, 4$, and 5 result in similar performance, indicating robustness and flexibility in selecting this parameter. For general POMDP problems, I recommend using the default value $k = 3$ (chosen arbitrarily before parameter experiments) or the optimal clustering value, obtained during the list initialization update before MCTS, which $k_i = 4$ in our paper. If the value $k_i = 4$ obtained in the list initial update before MCTS is used, this parameter $k$ becomes non-heuristic and consistently achieves good performance.

**Coefficients $\omega_1$ and $\omega_2$:**

These coefficients are straightforward to select. Without specific domain knowledge, I use the common value $\omega_1 = \sqrt{2}$, derived theoretically from the multi-armed bandit problem based on Hoeffding's inequality. For $\omega_2$, ensure that $\omega_2 range_i$ is comparable to the other two terms: $\hat{Q}(\boldsymbol{node}_o, \boldsymbol{a})$ and $\omega_1 \sqrt{\frac{\log N(\boldsymbol{node}_o)}{N(\boldsymbol{node}_o, \boldsymbol{a})}}$.

**Self-Defined Exploration Constant $C_r$:**

To determine $C_r$, users can follow this simple process:

- Run the POMDP problem for one step within the time limit and identify the number of their commonly used particles, like $N_p = 150$.
- Estimate the mean partitioning radius based on the problem setting, like $mean(range^*)$ is about 0.5.
- The selection of $C_r$ is to make sure that $1/(C_r mean(range^*)^2)$ is about 30%-50% of the particle number $N_p$. This ensures partitioning refines the continuous action domain at least 3-4 times.
- We have the selection $C_r$ is set as $1/((0.3 \text{ to } 0.5)N_p mean(range^*)^2)$.

**Coefficients for Refining Velocity $\bar{\omega}_2$ and $\bar{\omega}_1$:**

The coefficients $\bar{\omega}_2 = 0.3$ and $\bar{\omega}_1 = 0.6$ were chosen arbitrarily, without extensive consideration. Other similar settings should also work well.

---

[10]In fact, because the workspace is located in 4 areas, the suitable parameters $k$ should be equal to 4, which will be shown in Section K. Without loss of generality, we use 3 to get our main results, which still shows the dominant performance.

**Minimum Radius for Partitioning** $D_{lim}$:

In this paper, $D_{lim}$ is set to 0.2 without significant adjustments. This value covers a small region in $\mathbb{R}^9$, including 6D base pose, $p$, lift height $l_h$, and the pan and tilt angles $l_p$ and $l_t$ of the robot's head, as demonstrated with the Fetch robot. For example, two configurations with a distance of 0.2 might differ by 0.05 meters in $x$, $y$ axis, 0.1 radians (about 5.7 degree) in orientation, 0.05 meters in $l_h$, 0. 1 radians in $l_p$, 0. 1 radians in $l_t$. The resulting Euclidean distance is $\sqrt{(0.05)^2 + (0.05)^2 + (0.1)^2 + (0.05)^2 + (0.1)^2 + (0.1)^2} = 0.194$, which are within this small range. For other POMDP problems, users can adjust $D_{lim}$ to ensure actions within this range have similar physical meanings with acceptable differences.

## K  RESULTS FOR FETCH SIMULATOR WITH DIFFERENT PARAMETERS AND ABLATION STUDY

For our framework, the final performance of the proposed framework is commonly robust to the manual setting parameters. For example, the performance is nearly consistent, if orders of magnitude between $R_{max}$ and $R_{min}$ satisfy $R_{max} >> R_{min}$. When $R_{min}$ changes from -1 to -20, the final output performance is similar. We show the comparison results for the Hidden1 case in Table 3. This good robustness is inherited from the compact of the POMDP framework.

Table 2: 95% confidence interval of discounted cumulative reward, steps, and successful rate (within 50 steps)

| Scenarios | Hidden1 |
|---|---|
| $R_{min} = -1$ | $83377.1 \pm 6427.3 \mid 8.5 \pm 1.4 \mid 100\%$ |
| $R_{min} = -20$ | $84272.3 \pm 7170.9 \mid 8.8 \pm 1.9 \mid 100\%$ |

We also report very few cases in which parameters can significantly influence the method's performance, such as the threshold of declaring actions. This threshold governs the number of grid values used for comparison, which is crucial for determining the success of declared actions. The smaller $n_{odds}$ will make the task more challenging because we need to complete object detection from different orientations for each object. A smaller $n_{odds}$ makes the task more challenging, demanding object detection from various orientations for each object. To investigate the impact of this parameter, we conducted statistical analyses in Fig. 17 with $n_{odds}$ set to 2, 4, and 6 for several representative methods in a scenario featuring 6 objects (refer to Fig. 16). The successful rates for all methods in all these scenarios with different $n_{odds}$ are 100%. The advantage of our method reduces a lot with an easy $n_{odds}$ setting. In our final real world experiments using stretch robot, shown in Fig. 7, $n_{odds}$ is set as 6.

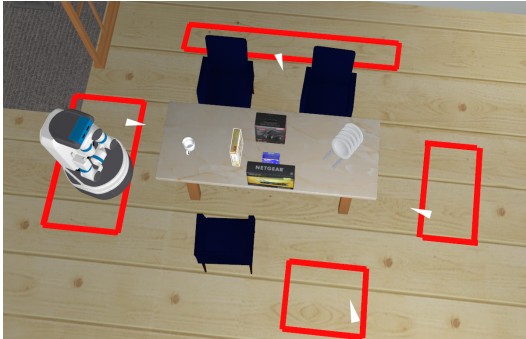

Figure 16: The scenario with 6 objects with candidate continuous position domain (red rectangles) and discrete pose domain (white triangles).

If we focus on our NPF-$k$CT method, the parameter $k$ is important to the refining speed of the continuous action domain and further decides the performance of the method facing different problems. In order to observe its effect, we change the clustering number $k$ from 2 to 8. For the scenario with 6 objects within Fig. 16, the comparison results about 95% confidence interval of the discounted cumulative rewards (Black line) and steps (Red line) are shown in Fig. 18. The results bounded in a colored dashed box are corresponding to the clustering number with the same color. We can find

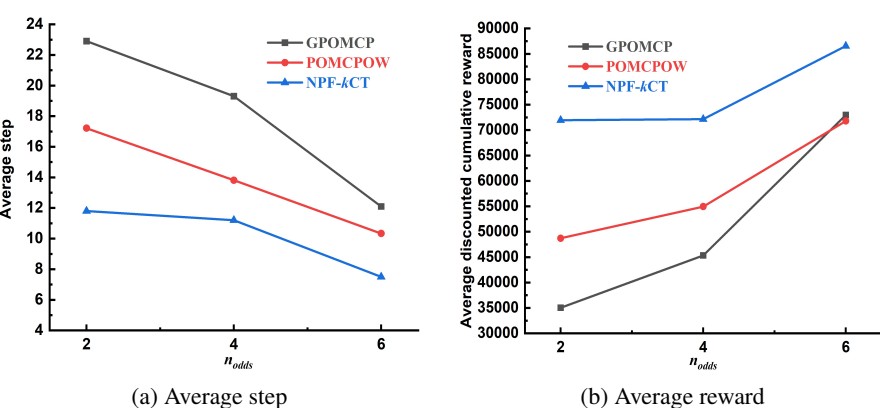

(a) Average step            (b) Average reward

Figure 17: Results for different $n_{odd}$ values.

that the most suitable number for this task is 4. Too large and too small clustering numbering will reduce the solver's performance.

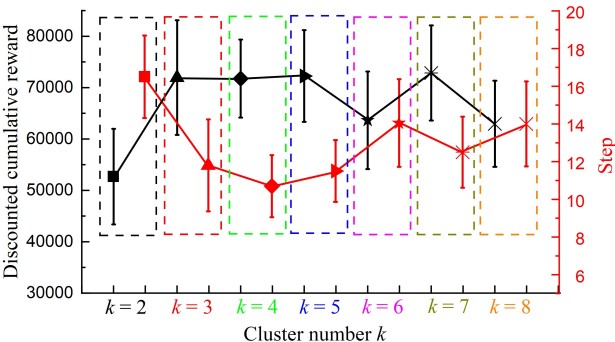

Figure 18: Results for different cluster numbers $k$.

Here, we would like to add experiments to complete the ablation study. It is easy to know that the neural process is definitely useful for the task, so we do not remove the neural process part and the main ablation is implemented for the k-center clustering. We remove the k-center clustering for refining the continuous action domain, which is described in Refine($\mathcal{T}$, $\boldsymbol{node}_o$, $\boldsymbol{a}^*$) in Algorithm 2 and Algorithm 3 and remain the action clustering and list initial update before MCTS to summarize the samples from neural process, named as NPF method. The comparison results for the Loose1 and Hidden1 cases are reported:

Table 3: Ablation study by removing function Refine($\mathcal{T}$, $\boldsymbol{node}_o$, $\boldsymbol{a}^*$)

| Scenarios | Loose1 | Hidden1 |
|---|---|---|
| NPF | $88653.8 \pm 6812.4 \mid 6.7 \pm 0.9 \mid 100\%$ | $75139.5 \pm 11521.5 \mid 11.4 \pm 2.7 \mid 100\%$ |

The result shows that the $k$-center clustering for refining the continuous action domain is useful and can improve the performance of our POMDP solver.

## L    RESULTS USING STRETCH ROBOT SIMULATOR AND SOME DETAILS IN REAL-WORLD EXPERIMENT

As mentioned before, our proposed method is not only limited to be used for the Fetch robot platform. Our NPF-$k$CT method can be transplanted to any mobile robot with the same sensor kinds and similar configuration. As an example, the Stretch robot simulator and its real-world platform are connected with our method to further verify the practice of the proposed framework. Even though the constructions of the Stretch robot and the Fetch robot are greatly different, we test the existing network pre-trained based on the Fetch simulator without recollecting the new data to evaluate the generalization ability of the scoring network in updating the grid belief using the head

camera. We follow all the problem settings shown in Appendix J, but the lift motion $l_h$ is removed in the observation function. Based on the Loose1 and Fig. 16 scenarios and the Stretch robot, the discounted cumulative reward, steps, and successful rate of the NPF-$k$CT method are $85578.6 \pm 8853.5 \mid 7.3 \pm 1.3 \mid 100\%$ (Loose1) and $52820.7 \pm 8146.1 \mid 17.6 \pm 4.2 \mid 100\%$ (Fig. 16). We can find that, due to the great configuration differences, the performance of the NPF-$k$CT method reduces a little compared with the results using the Fetch robot reported in Table 1 and Fig. 17. We recollect the data using the Stretch robot simulator, re-train the network with the same settings, and finally re-run the whole planning. The new results are $90233.7 \pm 4204.3 \mid 6.5 \pm 0.7 \mid 100\%$ (Loose1) and $63855.1 \pm 8922.9 \mid 14.0 \pm 2.7 \mid 100\%$ (Fig. 16), which shows similar performance than the result using the Fetch robot. A visual process of completing the object search task in 6 steps with the Stretch simulator for the Loose1 scenario is shown in Fig. 19.

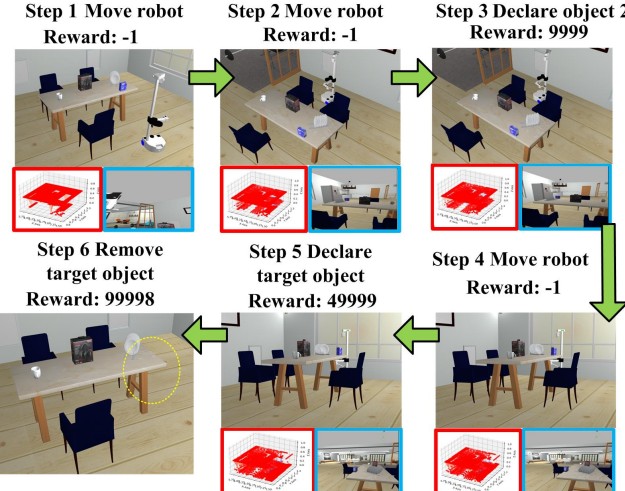

Figure 19: The visual progress for Loose1 scenario using the Stretch robot.

In the environment shown in Fig. 7, the 2D occupancy grid map and the point cloud map generated using RTAB-SLAM Labbé & Michaud (2019) are shown in Fig. 20 a and b for localization. In the navigation, we fuse these two maps to have a larger and safer map in Fig. 20 c and d.

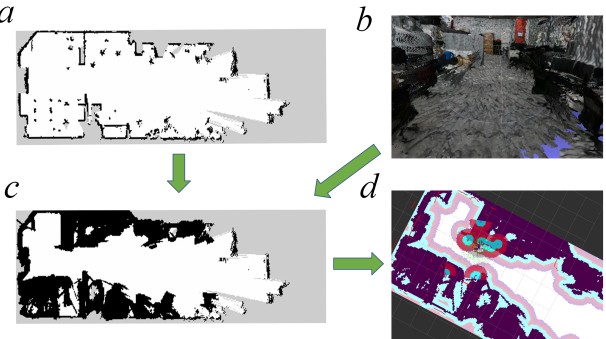

Figure 20: The used maps in real world experiments.

In the paper, we began to record 18 experiment results after the code demonstrated the ability to complete the picking task three consecutive times. Despite this, many errors were sporadic and could not be consistently replicated. Common communication and hardware issues included:

- **Mode-Switching Failures on the Stretch Robot**: The Stretch robot operates in three modes—navigation, trajectory, and position—but transitions between these modes were not always robust, leading to unresponsiveness. This issue arises from a bug in the Stretch robot's software, which does not launch all necessary drivers at the beginning. For example, navigation mode cannot use the arm, requiring a switch to position mode to activate joint drivers. These switches are not robust and occasionally fail.

- **ROS Waiting Time Errors**: At times, nodes experienced unexpected timeouts or synchronization issues, which were difficult to track as they did not occur consistently. In the rostopic waiting period, the sensor does not offer the message and then the code fails.

- **Delayed or Dropped Messages in ROS Communication**: I suspect this issue might be related to the communication setup. The Stretch robot is based on ROS2, my code is ROS1, and I use a ROS1-to-ROS2 bridge to enable communication between the systems. I think it is not very stable.

Such errors are inherent to real-world robotics experiments, particularly with ROS-based systems, and highlight the need for robust error-handling mechanisms. We are still improving it.

