# OpenReview forum: "NPF-$k$CT: A $k$-center clustering solver with neural process filter for continuous POMDP-based object search"
_ICLR.cc/2025/Conference — Submitted to ICLR 2025_

### Official Review · Reviewer_4caU · 2024-10-23

**Soundness:** 2
**Presentation:** 2
**Contribution:** 2
**Rating:** 3
**Confidence:** 4

**Summary:**

This paper proposes a POMDP solver to find the target occluded objects in complex household environments. The proposed method can select the appropriate action primitives and solve long-horizon goals, such as searching the target objects. The experiments are conducted in both simulation and the real world.

**Strengths:**

1: The authors work on an important research direction: searching for occluded objects is an interesting and open research problem.

2: The POMDP formulation is a feasible approach to solving the complex search problem.

3: The authors compare the proposed approach to several POMDP baselines, which is nice.

**Weaknesses:**

1: My first concern is that the paper does not discuss any failure cases. Throughout the paper, I find that the proposed approach achieves a 100% success rate in most of the simulation tasks and 10/18 in the real-world tasks. The authors only mention that the real-world failures are due to hardware and communication issues, which are superficial.

2: My second concern is that the paper misses an important discussion of related works on reasoning about occluded objects in the robot manipulation literature, such as [1][2]. Specifically, [2] proposes a memory model to remember and retrieve the occluded target objects by selecting action primitives to achieve the goal. I understand that both [1] and [2] only present experiments with a fixed-base arm, but it should be feasible to extend their approaches to mobile manipulation by incorporating navigation skills.

[1]: C. R. Garrett, C. Paxton, T. Lozano-P ́erez, L. P. Kaelbling, and D. Fox, “Online replanning in belief space for partially observable task and motion problems,” in 2020 IEEE International Conference on Robotics
and Automation (ICRA). IEEE, 2020.

[2]: Y. Huang, J. Yuan, C. Kim, P. Pradhan, B. Chen, L. Fuxin, and T. Hermans. Out of Sight, Still in Mind: Reasoning and Planning about Unobserved Objects with Video Tracking Enabled Memory Models. In IEEE International Conference on Robotics and Automation (ICRA), 2024.

3: The authors do not discuss any limitations of the proposed approach.

4: The visualizations in the paper can be significantly improved. For example, in Figure 7, it’s hard to understand the difference between the red boxes, and blue, and purple dashed circles without a caption. Furthermore, the approach section is difficult to follow without a clear approach overview figure.

5: This paper makes strong assumptions about known furniture and known object poses. In the real world, would your proposed approach be robust to the noisy segmentations and estimated object poses?

**Questions:**

1: what’s the difference between your proposed approach and [1][2]?

2: what are the failure cases and limitations of your proposed approach?

3: Could you show the performance of your proposed approach with imperfect segmentation and pose estimation in the real world?

---

> ### Author Response · Authors · 2024-11-23
> **Response to Reviewer 4caU for Submission185 (Part 1)**
>
> $\textbf{W1 W3 and Q2. My first...  The authors do not discuss... What are the failure...}$
>
> Answer: Thank you for your constructive comments. We appreciate the opportunity to address the concerns regarding failure cases and have added a detailed discussion in the paper to highlight the limitations and potential failure scenarios of our framework.
>
> In our Gazebo simulations, we utilize a self-defined robot equipped with highly accurate sensors to generate precise point cloud maps. Localization is initialized using Gazebo's true position data, ensuring accurate navigation. Even with this setup, small perception errors persist, such as incorrectly introducing additional objects for a single detected object. However, our framework is designed to be robust against such errors. Based on the assumption of a 100% execution success rate, as described in Footnote 4, no failure cases were observed in our experiments other than cases where the step limit was reached. The reported high success rate in simulation stems from this assumption: we presumed that primitive actions always succeed and implemented this via Gazebo's set_model_state function with added Gaussian noise following standard control operations. While certain actions—such as moving the base to a suitable pose, adjusting the lift height, or orienting the head—are inherently robust and achieve near 100% success, other actions, like object removal, present greater challenges. Without this assumption, the success rate of our framework in simulation would significantly decrease. Addressing this limitation will be a key focus of our future work.
>
> Unsuccessful execution may cause totally different challenges to our framework. Our current framework can solve some of them robustly. For example, if the robot fails to pick up an object and instead pushes it over, the framework can often recover by detecting the new object state, updating the belief accordingly, and continuing the task. While this may increase the number of steps required, it typically does not prevent the task from being successfully completed. However, certain failures in primitive actions can significantly impact overall performance. For example, an unsuccessful removal action that pushes two objects together can cause problems. Since our framework relies on point cloud segmentation to distinguish objects, such a failure could lead to incorrect bounding boxes, resulting in erroneous data association. This, in turn, can negatively affect belief updates and subsequent removal actions, ultimately causing the entire task to fail. Addressing these types of failure scenarios is an important direction for future improvements.
>
> To address poor primitive reliability, we propose several approaches:
>
> $\textbf{Incorporating Action Failure in Planning}$: Through prior experiments, we can estimate failure rates for various actions in specific scenarios and integrate these probabilities into the POMDP transition function. This would allow the planning process to account for action failures and mitigate their effects.
>
> $\textbf{Improving Primitive Success Rates}$: Another way is to improve the successful rate of the primitives using the following possible ways:
>
> 1. Action Execution Monitoring: Introduce robust monitoring systems to detect and respond to failures during primitive action execution. This could involve incorporating sensors or feedback mechanisms to validate the outcome of each action. If a failure is detected, the system can trigger a reattempt of the removing action or take corrective measures to ensure successful execution.
>
> 2. Enhancing Implementation Methods: Enhance action implementation capabilities using more robust and advanced methods to mitigate the impact of uncertainties on primitive execution. Like, the move_it toolbox may not be robust enough in the complicated environment based on the default setting. For example, we can adjust the used configurations in move_it to improve success rates in complex environments.
>
> 3. Robust Perception and Sensing: Enhance perception and sensing capabilities to mitigate the impact of uncertainties on primitive execution. By using more robust and diverse sensory inputs, the system can better adapt to changes in the environment and make informed decisions even in the presence of imperfect or incomplete information.
>
> Please refer to Part 2...

---

> ### Author Response · Authors · 2024-11-23
> **Response to Reviewer 4caU for Submission185 (Part 2)**
>
> In our view, the primary challenge of deploying our system to real-world robots lies in addressing errors and failures in perception, navigation, and execution, rather than in the planning component, which forms the core of our contribution. In our experiments, we carefully constructed the environment by selecting the most noise-free map and manually ensuring accurate localization initialization. Our real-world setting is relatively straightforward due to favorable conditions: objects were placed at relatively large distances (over 15 cm) from each other, and the target object was easily distinguishable from others due to its unique color (pure red) and shape (a bottle versus boxes). These factors minimized perception errors, which we did not observe significantly in our experiments. Addressing more complex and noisy real-world scenarios remains an important area for future work.
>
> $\textbf{W2. My second concern is that the paper misses...}$
>
> Answer: Thank you very much for your constructive comments. In response, we have updated the Related Work section by adding additional references focused on reasoning in mechanical search. We hope the revised version meets your expectations. Please refer to the updated Related Work section for details.
>
> $\textbf{W4. The visualizations in the paper can be ... }$
>
> Answer: Thank you for highlighting these visualization issues. We have updated the captions for Figure 7 (now Fig.6 in the revised version)  and the "Loose2" entry in Table 1 to provide clearer descriptions. Additionally, we add a new image to illustrate the step-by-step process of the proposed method in the supplementary materials. We hope the new visuals make it easier to understand and follow the steps of our proposed method. Please take a look and let us know if further clarification is needed.

---

> ### Author Response · Authors · 2024-11-23
> **Response to Reviewer 4caU for Submission185 (Part 3)**
>
> $\textbf{W5 and Q3. This paper makes strong assumptions about...  Could you show the performance...}$
>
> Answer: Thank you for your comments. To clarify, we do not assume prior knowledge of all object poses. What we intended to convey is that the objects included in the point cloud maps are known before the task begins. However, any additional objects not present in the initial point cloud must be detected online during the task. To improve clarity, we have revised the relevant sentences as follows:
>
> “we assume a pre-built point cloud and occupancy grid map, including furniture and some known objects, is available for planning. Other objects with unknown identities and poses are excluded.”
>
> When referring to "known furniture," I specifically mean large furniture pieces that define workspaces on their surfaces, such as tables, shelves, beds, and desks—not smaller items like chairs. If you agree that having access to a point cloud map of the environment is reasonable for common robotics planning tasks, you may also find it reasonable to consider known furniture. In typical living environments over a period of time (e.g., one month), large furniture is rarely moved, and the overall room structure remains unchanged. Therefore, before initiating a planning task, it is practical to use mapping techniques to generate a point cloud of the environment, including the large furniture and objects on its surfaces. This can be achieved using online semantic SLAM methods, such as [3], or by offline processes where the generated point cloud is converted into a semantic map through scene segmentation and object classification, as demonstrated in [4] and [5].
>
> That said, stronger navigation capabilities in fully unknown environments would certainly enhance the system. In future work, we aim to apply online semantic SLAM methods [3] to dynamically build navigation maps, identify workspaces, and execute tasks in real time.
>
> In this paper, we primarily focus on the planning component of the system, which constitutes the majority of the paper’s content. The perception aspect, which relies on established methods, serves as a supporting technology to ensure the overall process functions well under typical perception conditions. As such, its discussion in the main document is limited to a subsection in Section 3, with additional details provided in the supplementary material for ease of reproduction. Our framework is moderately robust to certain perception errors but not to large ones. For example, if a single object is mistakenly identified as two objects, the framework can handle this by introducing an additional substate into the state vector. While this may require more steps to confirm the object, the overall task can still be completed successfully. However, significant perception errors, such as object A being mislocated to the position of object B, would result in incorrect information being passed to the planning module, ultimately causing task failure.
>
> In summary, while our planning module can handle minor perception inaccuracies, it is not robust enough for highly challenging perception scenarios—a limitation that is common to other planning-focused works as well. Addressing these challenges remains an open area for future research.
>
> [3] Rosinol, A., Violette, A., Abate, M., Hughes, N., Chang, Y., Shi, J., Gupta, A. and Carlone, L., 2021. Kimera: From SLAM to spatial perception with 3D dynamic scene graphs. The International Journal of Robotics Research, 40(12-14), pp.1510-1546.
>
> [4] Zhao, Hengshuang, Li Jiang, Jiaya Jia, Philip HS Torr, and Vladlen Koltun. "Point transformer." In Proceedings of the IEEE/CVF international conference on computer vision, pp. 16259-16268. 2021.
>
> [5] Wu, Xiaoyang, Li Jiang, Peng-Shuai Wang, Zhijian Liu, Xihui Liu, Yu Qiao, Wanli Ouyang, Tong He, and Hengshuang Zhao. "Point Transformer V3: Simpler Faster Stronger." In Proceedings of the IEEE/CVF Conference on Computer Vision and Pattern Recognition, pp. 4840-4851. 2024.

---

> ### Author Response · Authors · 2024-11-23
> **Response to Reviewer 4caU for Submission185 (Part 4)**
>
> $\textbf{Q1. what’s the difference between your proposed approach and [1][2]?}$
>
> Answer: Thank you for your question. We have compared the proposed method with other approaches in multiple aspects, including the focused problem (scenario), problem formulation, proposed solvers, robot platforms used, and other settings. Due to the length limitations of the paper, these comparisons are not included in the main text but are addressed here.
>
> $\textbf{Focused problem: }$
>
> 1. NPF-kCT: This method addresses the problem of a robot searching for an object in a home environment with multiple rooms and workspaces. The robot has access to 3D point cloud and 2D occupancy grid maps, along with photos of the target object. This task involves planning under uncertainty with some physical interaction.
>
> 2. [1]: This approach focuses on planning in belief space for task and motion planning (TAMP) problems in a kitchen environment, such as cooking tasks. These tasks involve planning actions under uncertainty for complex activities that require physical interaction.
>
> 3. [2]: This method tackles reasoning and planning about unobserved objects using a video-based memory model to track object states and histories. It enables agents to consider unseen objects when planning.
>
> $\textbf{Built or used problem formulation:  }$
>
> 1. NPF-kCT: This method builds a self-defined POMDP formulation with a hybrid (continuous and discrete) action domain. The belief space includes the robot configuration, parameters of the guessed target object, and detected objects.
>
> 2. [1]: This approach formulates hybrid, belief-state Stochastic Shortest Path Problems (SSPP) using a POMDP framework with the PDDLStream framework, which uses predicate logic to describe planning problems. The formulation emphasizes computing the likelihood of observations.
>
> 3. [2] is to build a robot planning by finding skills, like push, pick-and-place, and pick-and-dump, and its parameters such that, when sequentially executed, transform the objects to satisfy all desired object and environment relations in the goal. The goal is defined as a logical conjunction of several desired objects and environment relations. It is not a POMDP formulation.
>
> $\textbf{Proposed solvers:}$
>
> 1. NPF-kCT: Proposes a novel online Monte Carlo Tree Search (MCTS)-based POMDP solver that integrates neural process filtering for primitive action selection, k-center clustering to refine the continuous action domain, and a revised UCB strategy for guiding belief tree exploration. This approach has the potential to be extended to other POMDP problems with hybrid action domains by replacing the neural process with other suitable action filter networks.
>
> 2. [1] is initially solved based on the classical method, the Focused algorithm, within domain-independent PDDLStream algorithms library and introduces online replanning by reusing previous deterministic plans. This represents a general framework for TAMP.
>
> 3. [2]: Introduces two novel methods—Latent Occluded Object Memory (LOOM) and Direct Occluded Object Memory (DOOM)—which integrate video tracker predictions with a relational dynamics model. These enable agents to consider unseen objects during reasoning and planning, offering a supportive strategy for TAMP with unobserved objects.
>
> $\textbf{Used robot platforms:}$
>
> 1. NPF-kCT: The default robot platform is a mobile manipulator equipped with an RGB-D camera and a 2D lidar. All perception modules are completed using onboard sensors. A single observation cannot cover the entire workspace, and objects may require operation from different grasp poses (not only top-down) with some being unreachable by the robot arm. Simulations are conducted in Gazebo using Fetch and Stretch robots. Real-world experiments use the Stretch robot.
>
> 2. [1]: The default platform is a wheeled robot arm with multiple degrees of freedom and an RGB-D camera fixed to the world frame. All perception is performed using fixed sensors, providing stable observations. A single observation covers the entire workspace, objects are usually operated using top-down grasps, and all objects are reachable by the robot arm. Simulations are conducted in PyBullet and PDDLStream using the Franka Panda robot. Real-world experiments also use the Franka Panda robot.
>
> 3. [2]: The default platform is a fixed robot arm with multiple degrees of freedom and an RGB-D camera fixed to the world frame. All perception is performed using fixed sensors, providing stable observations. A single observation covers the entire workspace, objects require operation from different grasp poses (not only top-down), and all objects are reachable by the robot arm. Simulations are conducted in PyBullet (Guessed by images) using the Kuka robot arm, and real-world experiments use the Kuka robot arm.
>
> Please refer to Part 5...

---

> ### Author Response · Authors · 2024-11-23
> **Response to Reviewer 4caU for Submission185 (Part 5)**
>
> $\textbf{Limitation:}$
>
> 1. NPF-kCT: Depends on the robust navigation, perception, and excution modules.
>
> 2. [1]: Computationally expensive for high-dimensional belief spaces.
>
> 3. [2]: Reliance on robust video tracking; limited to tasks involving object history.
>
> $\textbf{Key contribution }$
>
> 1. NPF-kCT: A novel online POMDP solver fusing MCTS, neural process filtering, and k-center clustering
>
> 2. [1]: A novel real-time replanning for failure execution of the primitive actions.
> 3. [2]: A novel video-tracking memory for reasoning with occluded objects, novel objects appearance, and object reappearance to complete TAMP.

---

> ### Comment · Area_Chair_acPG · 2024-11-25
>
> Dear Reviewer,
>
> Please provide feedback to the authors before the end of the discussion period, and in case of additional concerns, give them a chance to respond.
>
> Timeline: As a reminder, the review timeline is as follows:
>
> November 26: Last day for reviewers to ask questions to authors.
>
> November 27: Last day for authors to respond to reviewers.

---

> ### Comment · Reviewer_4caU · 2024-11-27
>
> Thank you to the authors for the detailed response. Part of my concerns have been addressed. Incorporating more related works and discussing limitations has improved the paper.
>
> However, the authors did not directly respond to my questions about failure cases. Instead, they attributed the failures to imperfect perception and action execution but did not provide any specific examples. In the simulation, if your current model consistently achieves an almost 100% success rate, I believe the tasks may be too easy. As a reader, I would like to understand under what circumstances your proposed approaches would fail. Specifically, I would like to know how your planning system fails. Simply attributing failures to perception and action execution is not convincing to me.
>
> For the real-world experiments, the authors mentioned "In 18 real-world trials, 10 were successful, and 8 failed due to hardware or communication issues." What specifically were the hardware and communication issues? No detailed answers were provided in either the initial version or during the revision period.
>
> In summary, while I agree that this paper addresses an important research direction, I have decided to maintain my original recommendation to reject the paper due to the remaining concerns outlined above.

---

> > ### Author Response · Authors · 2024-11-27
> > **We have updated the document about the hardware or communication issues**
> >
> > We have updated the document about the hardware or communication issues.

---

> > ### Author Response · Authors · 2024-11-27
> > **Response to Reviewer 4caU (Part 1)**
> >
> > Thank you very much for reviewing our rebuttal.
> > 1. For the comment: “but did not provide any specific examples.”
> >
> > Please refer to Response to Reviewer 4caU for Submission 185 (Part 1), where we provided an example of a real-world application where action execution failures occur:
> >
> > “For example, an unsuccessful removal action that pushes two objects together can cause problems. Since our framework relies on point cloud segmentation to distinguish objects, such a failure could lead to incorrect bounding boxes, resulting in erroneous data association. This, in turn, can negatively affect belief updates and subsequent removal actions, ultimately causing the entire task to fail. Addressing these types of failure scenarios is an important direction for future improvements.”
> >
> > 2. For the comment: “In the simulation, if your current model consistently achieves an almost 100\% success rate, I believe the tasks may be too easy.”,  we appreciate the reviewer’s observation regarding the high success rate of our model in simulations. However, we respectfully disagree with the conclusion that this implies the tasks are “too easy.” To clarify, consider the following thought experiment:
> >
> > In a real-world scenario, a robot searches for an object in several workspaces (limited areas) with perfect perception and action execution. The number of objects in the workspace (e.g., 5-10) is limited, and all objects are movable (since immovable surrounding objects would make the task unsolvable). Even using a random policy (any policy can cover the whole solution space probabilistically, like the UCB strategy includes a random part to explore), all these FOVs of the sequential robot poses can finally cover the workspace and find the target object with 100\% success rate by randomly changing its configuration and removing obstacles, if no step limit is imposed. This behavior is similar to a blanket search and guarantees success due to the finite nature of the problem. However, when the robot operates in a large workspace with many objects, the required steps increase significantly, while the 100% success rate is maintained. For example, in the complex1 scenario, our solver cannot always solve the problem within 50 steps. Importantly, the underlying planning problem based on POMDP belongs to the class of NP-hard problems (PSPACE-complete), which are inherently not simple. $\textbf{The 100 percent success rate arises from the inherent limitations of the workspace and the manageable number of objects, not because the problem is trivial.}$
> >
> > However, achieving 100% success with an unbounded number of steps is not practical for real-world applications, particularly in large-scale or complex environments where efficiency is critical. The goal of our work is to develop a method that minimizes the number of steps required, making it more suitable for real-world scenarios. Efficient solutions are crucial as task complexity scales up, for example, in larger action space or when dealing with more objects.
> >
> > In robotics, it is common practice to evaluate algorithms on small and medium-scale problems (our continuous action domain means that the problem scale is large) to establish a baseline for efficiency. These controlled setups allow us to compare different methods systematically and identify the most efficient approach for practical applications. We believe our approach demonstrates significant improvements in efficiency compared with SOTA POMDP methods, even for these foundational scenarios, and provides a solid basis for tackling larger and more complex problems in future work.
> >
> > 3. For “As a reader, I would like to understand under what circumstances your proposed approaches would fail. Specifically, I would like to know how your planning system fails. Simply attributing failures to perception and action execution is not convincing to me.”, as mentioned in Response to Reviewer 4caU for Submission 185 (Part 1) and above (Point 2), we did not observe failure cases with perfect perception and action execution when most objects were movable. Failures are primarily attributed to specific conditions such as perception noise or action execution inaccuracies, as discussed in the provided examples.

---

> > ### Author Response · Authors · 2024-11-27
> > **Response to Reviewer 4caU (Part 2)**
> >
> > 4. For comment “For the real-world experiments, the authors mentioned "In 18 real-world trials, 10 were successful, and 8 failed due to hardware or communication issues." What specifically were the hardware and communication issues? No detailed answers were provided in either the initial version or during the revision period.”
> >
> > If the reviewer has experience with ROS1 on a real mobile robot manipulator, you might be familiar with the challenge of code suddenly failing due to untraceable errors. In the paper, we began to record 18 experiment results after the code demonstrated the ability to complete the picking task three times. Despite this, many errors were sporadic and could not be consistently replicated.
> > Common issues included:
> >
> > (1) $\textbf{Mode-Switching Failures on the Stretch Robot:}$ The Stretch robot operates in three modes—navigation, trajectory, and position—but transitions between these modes were not always robust, leading to unresponsiveness. This issue arises from a bug in the Stretch robot’s software, which does not launch all necessary drivers at the beginning. For example, navigation mode cannot use the arm, requiring a switch to position mode to activate joint drivers. These switches are not robust and occasionally fail.
> >
> > (2) $\textbf{ROS Waiting Time Errors:}$ At times, nodes experienced unexpected timeouts or synchronization issues, which were difficult to track as they did not occur consistently. In the rostopic waiting period, the sensor does not offer the message and then the code fails.
> >
> > (3) $\textbf{Delayed or Dropped Messages in ROS Communication:}$ I suspect this issue might be related to the communication setup. The Stretch robot is based on ROS2, my code is ROS1, and I use a ROS1-to-ROS2 bridge to enable communication between the systems. I think it is not very stable.
> > Such errors are inherent to real-world robotics experiments, particularly with ROS-based systems, and highlight the need for robust error-handling mechanisms. We are still improving it.
> >
> > ... other questions. Many of them are hard to identify.
> >
> > I am adding a paragraph to show the error in the paper.  Please read the rebuttal first.

---

### Official Review · Reviewer_eZC5 · 2024-11-04

**Soundness:** 3
**Presentation:** 2
**Contribution:** 2
**Rating:** 6
**Confidence:** 4

**Summary:**

The paper presents an interesting approach to address the problem of Object Search. It uses the Monte-Carlo Tree Search (MCTS) along with a Neural Process Filter (NPF) and k-center clustering to find the best action based on maps, images and onboard sensors. The NPF helps remove the ineffective actions and the k-center clustering helps to refine the action spaces. This method has been tested in simulation on Gazebo and in real-life, showcasing some promising results.

**Strengths:**

- The paper is very well-written and structured.
- The paper presents a novel, simple yet elegant solution to the object search problem.
- The real-world demonstration of the method is appreciated.

**Weaknesses:**

1. The overall methodology might be novel but considering the individual modules, it comes down to a Monte-Carlo Tree Search (MCTS), an action scoring network (NPF) and k-center clustering. This approach seems heavily engineered towards solving the Object Search problem specifically and holds little appeal for the broader learning community.

2. Missing SOTA and comparitive study : The paper has not cited or evaluated against some known methods in the domain of Object search such as [1]. This makes it difficult to understand not just the comparative efficacy of the proposed method but also the complexity of the experimental setup.

3. The paper primarily uses the scenarios created by the authors through Gazebo for evaluation. However it would be better and more helpful to add experiments on some known dataset or benchmarks such as ScanNet [2].

4. The experimental setup seems too simple. In 4 out of the 5 scenarios the paper shows results in Table 1, the baselines and this method achieve well over 90% successful rate of object search within 50 steps. This warrants the need for more thorough evaluation either by increasing the complexity of the simulated setting or testing on some known dataset/benchmark as previously stated.

[1] Francesco Giuliari, Geri Skenderi, Marco Cristani, Alessio Del Bue, and Yiming Wang. 2023. Leveraging Commonsense for Object Localisation in Partial Scenes. IEEE Trans. Pattern Anal. Mach. Intell. 45, 10 (Oct. 2023), 12038–12049.\
[2] A. Dai, A. X. Chang, M. Savva, M. Halber, T. Funkhouser, and M. Nießner, “Scannet: Richly-annotated 3d reconstructions of indoor scenes,” in Proceedings of the IEEE/CVF Conference on Computer Vision and Pattern Recognition (CVPR), 2017.

**Questions:**

Same as Weakness

---

> ### Author Response · Authors · 2024-11-23
> **Response to Reviewer eZC5 for Submission185 (Part 1)**
>
> $\textbf{W1. The overall methodology might be novel but}$…
>
> Answer: Thank you for recognizing the novelty of our work and for providing valuable feedback. While our methodology is highly engineered for object search, its core contributions are broadly applicable beyond this specific task. Let us delve into the modular components—Monte Carlo Tree Search (MCTS), k-center clustering, and the action scoring network (NPF)—to highlight their individual roles and broader relevance.
>
> The combination of MCTS and k-center clustering provides a generalized approach for solving POMDPs with hybrid or continuous action spaces. MCTS is a well-established method for POMDP solvers, offering a structured way to explore and evaluate potential actions. By integrating k-center clustering, high-dimensional continuous action spaces can be discretized efficiently, enabling practical decision-making within these spaces. These two modules can operate together independently of the action scoring network, effectively functioning as a generalized POMDP solver applicable across various domains.
>
> Regarding NPF filtering, our goal is not to mandate the use of the same network for other POMDP tasks but rather to emphasize the action-filtering concept. A common approach in POMDP learning is to directly map beliefs to policies, aiming for long-term objectives such as maximizing discounted cumulative rewards, as seen in Q-learning networks [3]. However, this approach is notoriously challenging and often underperforms due to the extremely large minimum description length [4], especially when dealing with high-dimensional belief spaces and complex models. Our method simplifies this by organically integrating the NPF scoring network with a compact POMDP solver. The NPF scoring network prioritizes short-term, physically meaningful goals, making it significantly easier to train. This network efficiently filters out ineffective actions, streamlining the decision-making process. We have clarified this motivation in the revised text. Importantly, while the specific NPF structure is task-specific, introducing networks focused on short-term goals within POMDP solvers is a broadly applicable concept that can benefit the wider learning community.
>
> In short, we are recommending a network filtered POMDP solver for general POMDPs with hybrid action domain and complicated primitives.
>
> [3] Hausknecht, Matthew, and Peter Stone. "Deep recurrent q-learning for partially observable mdps." In 2015 aaai fall symposium series. 2015.
>
> [4] Zirui Zhao, Wee Sun Lee, and David Hsu. Large language models as commonsense knowledge for large-scale task planning. Advances in Neural Information Processing Systems, 36, 2024b.

---

> ### Author Response · Authors · 2024-11-23
> **Response to Reviewer eZC5 for Submission185 (Part 3)**
>
> $\textbf{W3. The paper primarily uses the scenarios created by the authors through Gazebo for evaluation. However it would be better}$ $\textbf{and more helpful to add experiments on some known dataset or benchmarks such as ScanNet [2].}$
>
> Answer: Thank you very much for your valuable comments. Reasoning tasks in mechanical search typically do not require physical interaction with the environment. The process involves providing a built environment or partial scene observations, such as point clouds, and using high-level cognitive reasoning to predict the position of the target object. These predictions are then compared to ground truth data, followed by statistical analysis to evaluate performance. Once the prediction is complete, the task is considered finished. As a result, reasoning tasks often rely on well-established public datasets and benchmarks.
>
> In contrast, our focus is on the robotics planning aspect of mechanical search, which is fundamentally different from reasoning tasks in artificial intelligence. Robotics planning requires active interaction with the environment, such as removing or rearranging objects using a robotic arm. This type of task necessitates at least a simulated environment with a physical engine, such as Gazebo. Therefore, point cloud datasets like ScanNet [2] are not suitable for our problem.
>
> In the planning domain, there is no universal dataset specifically tailored to mechanical search because tasks are highly dependent on the robotic platform, task goals, and environment. Researchers often generate custom datasets or scenarios using simulation environments like AI2-THOR, PyBullet, or Fetch/Stretch simulators to meet their specific needs. Each lab has different robot platforms, and their lab settings are different. The variability of robotic platforms further complicates standardization, as different robots have distinct action domains. For example, the Fetch robot can move its lift, whereas the Stretch robot cannot.
>
> The state-of-the-art references in the planning domain mentioned in our paper illustrate this diversity:
>
> 1. [6] uses an ABB YuMi robot with suction-cup and parallel-jaw grippers, focusing on tasks within a bin (box).
>
> 2. [7] involves a Fetch robot with a stationary base, limited to scenarios within a box-like shelf with one open side.
>
> 3. [8] employs robots like Boston Dynamics Spot and Kinova MOVO, which do not use robotic arms to interact with the environment and do not address occlusion relationships.
>
> 4. [9], similar to our work, uses a Fetch robot in the Gazebo environment and interacts with the environment, such as removing objects, making it a suitable comparison for our method.
>
> In summary, mechanical search tasks in the planning domain are highly platform-dependent and task-specific, which limits the development of standardized datasets. Public datasets are often impractical unless the exact platform and task goals align. To address this, we propose open-sourcing our code to enable other researchers to test and adapt it for their tasks. This follows the approach of other methods we compared against, such as POMCP, GPOMCP, POMCPOW, and VOMCPOW.
>
> [6] Danielczuk, Michael, Andrey Kurenkov, Ashwin Balakrishna, Matthew Matl, David Wang, Roberto Martín-Martín, Animesh Garg, Silvio Savarese, and Ken Goldberg. "Mechanical search: Multi-step retrieval of a target object occluded by clutter." In 2019 International Conference on Robotics and Automation (ICRA), pp. 1614-1621. IEEE, 2019.
>
> [7] Huang, Huang, Marcus Dominguez-Kuhne, Vishal Satish, Michael Danielczuk, Kate Sanders, Jeffrey Ichnowski, Andrew Lee, Anelia Angelova, Vincent Vanhoucke, and Ken Goldberg. "Mechanical search on shelves using lateral access x-ray." In 2021 IEEE/RSJ International Conference on Intelligent Robots and Systems (IROS), pp. 2045-2052. IEEE, 2021.
>
> [8] Zheng, Kaiyu, Anirudha Paul, and Stefanie Tellex. "A System for Generalized 3D Multi-Object Search." In 2023 IEEE International Conference on Robotics and Automation (ICRA), pp. 1638-1644. IEEE, 2023.
>
> [9] Chen, Yongbo, and Hanna Kurniawati. "POMDP planning for object search in partially unknown environment." Advances in Neural Information Processing Systems 36 (2024).

---

> ### Author Response · Authors · 2024-11-23
> **Response to Reviewer eZC5 for Submission185 (Part 4)**
>
> $\textbf{W4. The experimental setup seems too simple}$...
>
> Response: Thank you for your kind comments. However, we do not totally agree with this comment. In the planning domain of mechanical search, the problems we address are not simple. The high success rate (over 90%) we achieved is due to an assumption mentioned in Footnote 4: we assume that the primitive actions have a 100% success rate by leveraging the Gazebo server’s set_model_state function with added Gaussian noise after standard control operations. Without this assumption, our method would not achieve such a high success rate. Addressing this limitation will be a focus of our future work. As emphasized in the Introduction, our primary contribution lies in task-level planning, not in designing optimized object-picking methods for primitive actions. We have also added a dedicated section discussing the limitations of our framework to clearly address this point.
>
> Our scenario is challenging for several reasons:
>
> 1. Interaction with the Environment: The planning environment is dynamic, requiring the robot to modify it actively.
> 2. Onboard Sensors: Perception must be completed without global information, introducing significant challenges.
> 3. Robot Base Movement: Moving the robot base introduces large perception errors (current localization methods based on AMCL can result in errors of approximately 10 cm, which is comparable to the size of many target objects).
>
> In contrast, most state-of-the-art (SOTA) methods, such as [6] and [7], consider fixed-base robot platforms with global observation systems, often using external fixed RGBD cameras. These setups allow stable perception with minimal noise. For example, most industrial robots, such as robotic arms, typically do not move their base to maintain robustness. Consequently, our scenario is significantly harder than those in [6] and [7].
>
> Other SOTA methods, such as [8] and [10], involve moving bases and onboard sensors but do not account for interaction with the environment. Instead, they focus only on reconfiguring the robot itself without altering the external scene. This omission makes their scenarios less complex than ours. Additionally, these methods primarily use standard solvers like POMCP. Meanwhile, they are based on Object-Oriented POMDP (OO-POMDP), which is very popular, considers the state and observation spaces to be factored by a set of n objects, where each belongs to a class with a set of attributes, and the beliefs about different objects are independent for the small computational reason. We follow the same Object-Oriented idea because of the good applicability of the real-world application but go forward one step to consider the occlusion relationship for the belief update between different objects in a complicated 3D environment, including some challenging cases with partial and fully occluded objects.
>
> Regarding datasets and benchmarks, please refer to our previous response.
>
> In summary, our approach addresses a more challenging and dynamic setting compared to existing methods, highlighting the novelty and complexity of our work.
>
> [10] Zheng, Kaiyu, Rohan Chitnis, Yoonchang Sung, George Konidaris, and Stefanie Tellex. "Towards optimal correlational object search." In 2022 International Conference on Robotics and Automation (ICRA), pp. 7313-7319. IEEE, 2022.

---

> ### Author Response · Authors · 2024-11-23
> **Response to Reviewer eZC5 for Submission185 (Part 2)**
>
> $\textbf{W2. Missing SOTA and comparative study}$ ...
>
> Answer: Thank you very much for your constructive comments. In response, we have added more state-of-the-art (SOTA) references on reasoning to the Related Work section, as two reviewers expressed significant interest in the reasoning aspect of mechanical search. The reviewer’s suggested method, which leverages the Directed Spatial Commonsense Graph (D-SCG) to infer relative positions of objects using commonsense knowledge, is indeed an intriguing and promising direction. This approach aligns with our ongoing project exploring the use of hallucinated human-like reasoning [5] and commonsense to predict target object positions. However, they are more focused on the reasoning aspect of the object search problem rather than the planning and interaction aspect commonly emphasized in robotic systems. As a reasoning focused work, this work centers on predicting the location of the target object based on incomplete information (i.e., a partial 3D scan of the scene) and commonsense knowledge and emphasizes inferring spatial relationships and utilizing commonsense reasoning to localize/predict objects without direct interaction with the physical environment. In contrast, our work falls squarely within the domain of robotics, emphasizing planning and interaction during object search. Our approach involves strategies for physical navigation, managing occlusions, and actively modifying the environment to uncover hidden objects—key aspects not addressed in the aforementioned reasoning-focused work. This is not addressed in this paper.
>
> In summary, the two approaches address different problems. In reasoning-focused tasks, the goal is to predict the target object's position using incomplete information and commonsense knowledge, completing the task once the prediction is made. In contrast, planning in robotics involves an interactive process where the system physically navigates the environment, overcomes obstacles such as occlusions, and modifies the scene if necessary to locate the object. Planning not only predicts the object's position but also ensures it is physically found, emphasizing dynamic interaction with the environment.
>
> We have come to realize that the learning community shows a stronger interest in reasoning tasks. Thank you once again for this very constructive feedback. In response, we have updated the Related Work section by incorporating additional references related to reasoning in mechanical search. The updated section is as follows:
>
> “$\textbf{Mechanical Search Reasoning Methodologies:}$ In earlier robotics approaches, reasoning was often integrated into the planning process, such as updating probabilities, rather than being treated as a separate module. As robotic systems grew more complex, reasoning emerged as an independent layer, focusing on high-level cognitive tasks such as commonsense inference, contextual understanding, and hypothesis generation, which then guide a planning module to execute detailed action strategies. The work Giuliari et al. (2023) demonstrates a reasoning module to infer plausible object locations using environmental context and commonsense knowledge, aiding object localization in partially observed scenes. In more recent work  Ge et al. (2024), the authors utilize commonsense knowledge from large language models (LLMs) to construct scene graphs, enhancing object search in household environments. However, these commonsense-based approaches struggle with unconventional object arrangements, like a random setting. In our scenario, objects are placed to violate human habit, increasing search complexity and necessitating active robot-environment interaction, such as removing occlusions. Our reasoning idea is involved in the probability update of the grid world. By representing the belief over the pose state of each object in the planning environment using particle filtering, the authors in  Garrett et al. (2020) incorporate probabilistic reasoning into a deterministic planner and then complete the re-used replanning when facing the base movement failure. A recent work  Huang et al. (2024) leverages a video tracking-based memory model with reasoning and planning capabilities, allowing the system to remember the potential locations of the occluded target objects and complete tasks by selecting action primitives.”
>
> [5] Jiang, Yun, Hema Koppula, and Ashutosh Saxena. "Hallucinated humans as the hidden context for labeling 3d scenes." In Proceedings of the IEEE Conference on Computer Vision and Pattern Recognition, pp. 2993-3000. 2013.

---

> ### Comment · Area_Chair_acPG · 2024-11-25
>
> Dear Reviewer,
>
> Please provide feedback to the authors before the end of the discussion period, and in case of additional concerns, give them a chance to respond.
>
> Timeline: As a reminder, the review timeline is as follows:
>
> November 26: Last day for reviewers to ask questions to authors.
>
> November 27: Last day for authors to respond to reviewers.

---

> ### Author Response · Authors · 2024-11-29
> **Reminder and Appreciation to Reviewer eZC5**
>
> Dear Reviewer eZC5,
>
> We sincerely appreciate your time and effort in reviewing our work.
>
> As the discussion period approaches its conclusion, we would like to kindly remind you that only a few days remain for any additional comments or questions.
>
> In response to your feedback, we have recently provided a detailed reply addressing your concerns.
>
> Thank you once again for your valuable insights and thoughtful contributions.
>
> Best regards,
> The Authors

---

### Official Review · Reviewer_zokC · 2024-11-04

**Soundness:** 3
**Presentation:** 3
**Contribution:** 3
**Rating:** 8
**Confidence:** 3

**Summary:**

This paper introduces NPF-kCT, an online POMDP solver designed for efficient object search tasks in complex 3D environments using mobile manipulators. The paper outlines that the major challenges are searching for target objects in cluttered and partially known environments, where perception errors, limited fields of view, and visual occlusions are prevalent. NPF-kCT formulates the object search problem as a high-dimensional POMDP with hybrid action spaces and utilizes a neural process-based network to predict the feasibility of primitive actions and filter out ineffective ones. The method also applies a clustering method to group filtered actions into hyperspheres and employs MCTS in conjunction with a UCB strategy to construct a belief tree and select optimal actions. The paper provides theoretical results for sampling robot actions and empirical results in Gazebo simulations with Fetch and Stretch robot simulators that outperform baseline methods in terms of success rate and efficiency in finding target objects.

**Strengths:**

The results have a sufficient number of baselines. The work tackles a significant problem as open-world mechanical search is difficult and useful. The introduction and conclusion are clear but section 4 could do a better job with intuition about aspects of the method. The figures are good and informative but Figure 4 should be redone. The main novelty is the POMDP solver which has both empirical and theoretical results thus having good originality.

**Weaknesses:**

The paper only offers simulation experiments in Gazebo with two mobile robots but would benefit from expanding to other navigation simulators such as one of the THOR variants but ideally real experiments should be conducted to truly corroborate results. It would be interesting to see how this method scales with higher-dimensional belief states and action spaces. Line 246 does not seem to be a full sentence. The clarity for the Motivation about Network Filtering is still not obvious and would benefit from being clearer. The related works would also benefit from being more thorough. It would also be helpful to conduct ablations of different parts of the method.

**Questions:**

To solve the task of mechanical search, why is better to favor a POMDP solver rather than a VLA (vision-language-action) model?
How much benefit is the action clustering providing?

---

> ### Author Response · Authors · 2024-11-23
> **Response to Reviewer zokC for Submission185 (Part 1)**
>
> $\textbf{S: but section 4 could do a better job with intuition about aspects of the method. The figures are good and informative but Figure 4 should be redone.}$
>
> Answer: Thank you very much for your insightful comments in the Strength section. We have added a sentence to emphasize our intuitive idea in applying the neural process in short term (one step) learning and combine with POMDP solver: “Rather than training a complex network for long-term goals, as seen in Q-learning  Hausknecht & Stone (2015, with extremely large minimum description length  Zhao et al. (2024b) to pursue $\mathcal{X}={a}^*$, we use a simpler scoring network focused on short-term (even one step), physically meaningful goals. This approach filters out ineffective actions, allowing a compact POMDP to select the optimal action based on current beliefs.” Additionally, we have revised Figure 4 (now Fig. 3 in the updated submission) to clearly illustrate the refining actions. We hope these updates meet your expectations and address your concerns.
>
> $\textbf{W1: The paper only offers simulation experiments in Gazebo with two mobile robots but would benefit from expanding to other navigation simulators}$ $\textbf{such as one of the THOR variants but ideally real experiments should be conducted to truly corroborate results.}$
>
> Answer: Thank you very much for this professional comment. We agree that, compared to Gazebo, THOR variants are better suited for real-world validation and deployment testing as they account more accurately for actual physical conditions. Our project is still in development, and some platforms used in this paper are based on ROS 1, which makes Gazebo a more practical choice for rapid development and initial algorithm testing. Additionally, for generating machine learning data, Gazebo is more efficient and repeatable, allowing faster simulations compared to physical THOR variants. In the future, as our algorithm becomes more stable, we plan to incorporate THOR variants for extensive real-world testing.
>
> $\textbf{W2: It would be interesting to see how this method scales with higher-dimensional belief states and action spaces.}$
>
> Answer:  Thank you very much for your constructive comments. Our state space is defined to include all the detected objects, corresponding to a dynamically growing state space. In the Complex1 scenario with 15 objects, the state space can be considered as a high-dimensional case. For the action spaces, due to the degree of freedom of the robot platforms, we have considered all the available mechanical actions and some logical operations for the Fetch and Stretch robots, including base motion, lift motion, head motion, declaring action, and robot arm removing action. In the future, we will apply our NPF-kCT method to the problem with higher-dimensional belief states and more complex action spaces.
>
> $\textbf{W3. Line 246 does not seem to be a full sentence.}$
>
> Answer: Thank you very much for your comments. This point was also raised by other reviewers. The explanation beginning on Line 246 is intended to provide further details on specific functions and steps used within Algorithm 1. The phrase “where Net.sam(·) is the function that generates…” treats the algorithm as an object, with the subsequent lines elaborating on each element in the algorithm. This structure helps readers refer back to the algorithm while gaining a deeper understanding of its functionality. To enhance clarity, we have revised this section as follows:
>
> Replace “Where” by “Algorithm~1 includes functions and parameters that are detailed below:”
>
> $\textbf{W4. The clarity for the Motivation about Network Filtering is still not obvious and would benefit from being clearer.}$
>
> Answer: We have added several sentences to clarify the motivation about network filtering:
>
> “Rather than training a complex network for long-term goals, as seen in Q-learning  Hausknecht & Stone (2015, with extremely large minimum description length  Zhao et al. (2024b) to pursue $\mathcal{X}={a}^*$, we use a simpler scoring network focused on short-term (even one step), physically meaningful goals. This approach filters out ineffective actions, allowing a compact POMDP to select the optimal action based on current beliefs.”
>
> $\textbf{W5. The related works would also benefit from being more thorough.}$
>
> Answer: Thanks for your constructive comments. We have updated the related works by adding more references about the reasoning of the mechanical search. We hope the revised version meets your expectations. Please refer to the updated Related Works section for details.

---

> ### Author Response · Authors · 2024-11-23
> **Response to Reviewer zokC for Submission185 (Part 2)**
>
> $\textbf{W6. It would also be helpful to conduct ablations of different parts of the method. and Q2 How much benefit is the action clustering providing?}$
>
> Answer: We have added experiment to complete the ablation study of the method. It is easy to know that the neural process is definitely useful for the task, so we do not remove the neural process part. We focused the ablation on the k-center clustering used to refine the continuous action domain. In this modified setup, we remove the k-center clustering for refining the continuous action domain and retain the action clustering and list initial update before MCTS to summarize the samples from neural process, named as NPF method. The comparison results for the Loose1 and Hidden1 scenarios are as follows:
>
> Loose1: 88653.8±6812.4 | 6.7±0.9 | 100%
>
> Hidden1: 75139.5±11521.5 | 11.4±2.7 | 100%
>
> These results demonstrate that k-center clustering significantly enhances the performance of the POMDP solver by effectively refining the continuous action domain.
>
> $\textbf{Q1. To solve the task of mechanical search, why is better to favor a POMDP solver rather than a VLA (vision-language-action) model?}$
>
> Answer: Thank you very much for this interesting comments. Regarding the mechanical search task, we chose to use a POMDP framework for the following reasons:
>
> POMDPs excel in systematically handling uncertainty and partial observability. They are particularly well-suited for tasks requiring sequential decision-making under incomplete information, which is critical for mechanical search. In such tasks, actions must not only address current observations but also contribute to a long-term strategy that reduces uncertainty and maximizes the probability of finding the target. In my mind, the VLA is more efficient in short term task to mechanical search tasks within complicated visual scenarios, especially when there are clear visual and semantic cues guiding the search. Like [1]. This work can offer good semantic distributions based on the current visual RGB images for later Mechanical Search Policies, but it does not offer a theoretical and strategic sequence framework for long term target. Other modules need to be introduced for VLA to make exploration more thoroughly.
>
> For small labs, developing efficient POMDP models tailored to specific robotic platforms is feasible by carefully defining the transition model, observation model, and reward function. Such frameworks can be adapted to different environments by updating these models of uncertainty and dynamics. In contrast, while VLAs, such as CLIP, Flamingo, or RT-2, are gaining popularity, they require significant computational resources (e.g., GPUs) and extensive robotics datasets for training and deployment. This makes them challenging for smaller labs to adopt without access to these resources. While leveraging pre-trained VLA models and fine-tuning them for specific applications is possible, their broad design often limits customization for nuanced tasks. Consequently, reliance on these models may not always be practical for labs with limited computational or dataset resources, especially for tailoring solutions to specialized mechanical search problems.
>
> [1] Sharma, Satvik, Huang Huang, Kaushik Shivakumar, Lawrence Yunliang Chen, Ryan Hoque, Brian Ichter, and Ken Goldberg. "Semantic mechanical search with large vision and language models." arXiv preprint arXiv:2302.12915 (2023).
>
> Of course, we recognize the inherent limitations of the POMDP framework for mechanical search, particularly its challenges with large state/action spaces and its limited adaptability to high-level semantic reasoning or tasks requiring natural language understanding. However, for our specific platform, such as the Fetch robot, the action space is relatively manageable. Additionally, we can leverage powerful networks like YOLO to efficiently handle the perception component. To address these limitations and stay aligned with current advancements, we are also exploring a complementary project aimed at integrating large language models (LLMs) as a source of common knowledge for mechanical search. This approach allows us, as a small lab, to tap into cutting-edge developments while maintaining a practical focus on our resource constraints.

---

> > ### Comment · Reviewer_zokC · 2024-11-26
> >
> > Thanks for addressing my concerns, I have marginally improved my score.

---

> ### Comment · Area_Chair_acPG · 2024-11-25
>
> Dear Reviewer,
>
> Please provide feedback to the authors before the end of the discussion period, and in case of additional concerns, give them a chance to respond.
>
> Timeline: As a reminder, the review timeline is as follows:
>
> November 26: Last day for reviewers to ask questions to authors.
>
> November 27: Last day for authors to respond to reviewers.

---

> ### Author Response · Authors · 2024-11-26
> **Thanks for your reply.**
>
> Thank you for your response and for considering our rebuttal. We appreciate your effort in revisiting your review. However, we noticed that the score in the system has not changed (still 6). Could you kindly double-check whether this might be due to a system error or another issue? Thanks again for your help.

---

> ### Author Response · Authors · 2024-11-26
> **Thank you for updating score**
>
> Thank you very much for updating score from 6 to 8.

---

### Official Review · Reviewer_4TPu · 2024-11-07

**Soundness:** 3
**Presentation:** 2
**Contribution:** 3
**Rating:** 6
**Confidence:** 3

**Summary:**

This paper proposes an online POMDP solver, Neural Process Filtered k-Center Clustering Tree (NPF-kCT),  designed to optimize object search tasks for mobile robots in cluttered environments. NPF-kCT integrates Monte Carlo Tree Search (MCTS) with a neural process network and a k-center clustering approach, making action selection more efficient by filtering out sub-optimal actions from high dimensional action spaces.  Experiments in Gazebo and real-world settings show the solver's effectiveness, as it outperforms baseline POMDP methods in success rate, speed, and total reward for the same computation budget.

**Strengths:**

1. One of the primary drawbacks of tree search based algorithms is increasing sample/time complexity with larger tree widths. In this paper, NPF-kCT effectively combines neural process filtering with k-center clustering and MCTS to filter out sub-optimal actions while expanding the search tree.

2. Results include both simulated and real-world scenarios of object search in cluttered environments and the proposed method can generalize to different robot platforms. NPF-kCT especially outperforms comparable MCTS-based baseline algorithms in terms of number of actions or steps required to achieve the task.

**Weaknesses:**

1. NPF-kCT relies on a number of heuristic design choices which might hinder generalization and reproducibility. Some of these are setting the number of clusters $k$, number of simulation episodes in the search tree, and neural process network training which affects the filtered action space.

2. Certain assumptions, such as a 100% success rate for actions in Gazebo, may not fully reflect real-world constraints, possibly affecting the method’s performance in practical applications.

3. In terms of writing quality, although the authors have provided adequate details for the algorithm, it might help to make the text more succinct to further improve the presentation.

**Questions:**

1. In Table 1, do the reward values convey anything other than "higher reward is better"? Why not just report the number of steps since that more clearly supports the claim of improved search efficiency over baselines?

2. Line 246 - why does it start from the middle of a sentence?

3. Why was the time complexity not reported for any of the algorithms in Table 1? Is it efficient enough to actually use NPF-kCT as an online solver with real robots?

---

> ### Author Response · Authors · 2024-11-23
> **Response to Reviewer 4TPu for Submission185 (Part 1)**
>
> $\textbf{W1. NPF-kCT relies on...}$
>
> Answer: Thank you for pointing this out. For our method, like most MCTS methods, we do not give a fixed number of simulation episodes in the search tree. Our method is an anytime method. We give a limited planning time (60 seconds for each step in our paper) and keep running the episodes until the planner reaches the time limitation. Therefore, the number of simulation episodes is not a heuristic design choice. A key contribution of this paper is the neural process network, which is specifically designed for our focused task but can be extended to other POMDP problems. Instead of directly training a complex network to handle long-term goals (as seen in approaches like Q-learning, which suffer from high minimum description length), we propose a simpler scoring network for short-term goals with physical meanings. This approach makes training easier, helps filter useless actions, and allows the POMDP framework to identify optimal actions based on current beliefs. Hence, we think that the idea of using a neural process filter in POMDP is more important than the heuristic task-focused training. For example, consider a robot cooking a meal. Training a full Q-learning network to plan all necessary actions would be highly complex. Instead, our method can train a network to sample effective primitive actions, such as adjusting the angle of a robotic arm to pour water into a pot, and then use the POMDP framework to sequence these actions into successful cooking steps. This illustrates the generalizability of our method to other POMDP problems. Finally, to ensure reproducibility, we will open-source our code, allowing readers to replicate the results presented in this paper.
>
> For all the other parameters used in our solver, we explain them one by one. Please refer to Appendix J. Our solver relies on 7 parameters, including the refining clustering number $k$, coefficients $\omega_1$ and $\omega_2$ for the MCTS action selection strategy, a self-defined constant $C_r$, the minimum radius for partitioning $D_{lim}$, and two coefficients controlling the refining velocity, with $0 \leq \bar{\omega}_2 < \bar{\omega}_1 \leq 1$. Most of these parameters were not fine-tuned for optimal performance; instead, they were quickly identified or chosen intuitively. Here, we would like to help the users to determine the parameters within a short time, like 20 minutes.
>
> $\textbf{Refining Clustering Number $k$:}$ Fig. 18 show that $k=3, 4, 5$ result in similar performance, indicating robustness and flexibility in selecting this parameter. For general POMDP problems, I recommend using the default value $k=3$ (chosen arbitrarily before parameter experiments) or the optimal clustering value, obtained during the list initialization update before MCTS, which $k_i=4$ in our paper. If the value $k_i=4$ obtained in the list initial update before MCTS is used, this parameter $k$ becomes non-heuristic and consistently achieves good performance.
>
> $\textbf{Coefficients $\omega_1$ and $\omega_2$:}$ These coefficients are straightforward to select. Without specific domain knowledge, I use the common value $\omega_1=\sqrt{2}$, derived theoretically from the multi-armed bandit problem based on Hoeffding's inequality. For $\omega_2$, ensure that $\omega_2range_i$  is comparable to the other two terms: $\hat Q ({node}_o, {a})$ and $\omega_1 \sqrt{\frac{\log N ({node}_o)}{N ({node}_o,~{a})}}$.
>
> $\textbf{Self-Defined $C_r$:}$ Users can follow this simple process:
> 1. Run the POMDP problem for one step within the time limit and identify the number of their commonly used particles, like $N_p=150$.
> 2. Estimate the mean partitioning radius based on the problem setting, like $mean(range^*)$ is about 0.5.
> 3. The selection of $C_r$ is to make sure that $1/(C_rmean(range^*)^2)$ is about 30\%-50\% of the particle number $N_p$.  This ensures partitioning refines the continuous action domain at least 3-4 times.
> 4. We have the selection $C_r$ is set as $1/((0.3-0.5)N_pmean({range^*})^2)$.
>
> $\textbf{Refining Velocity $\bar{\omega}_2$ and $\bar{\omega}_1$:}$ $\bar{\omega}_2 =0.3$ and $\bar{\omega}_1=0.6$ were chosen arbitrarily, without extensive consideration. Other similar settings should also work well.
>
> $\textbf{Minimum Radius $D_{lim}$:}$ $D_{lim}$ is set to 0.2 without significant adjustments. This value covers a small region in $\mathbb{R}^9$, including 6D base pose, $p$, lift height $l_h$, and the pan and tilt angles $l_p$ and $l_t$ of the robot’s head, as demonstrated with the Fetch robot. For example, if two configurations with a distance of 0.2 might differ by 0.05 meters in $x$, $y$ axis, 0.1 radians in orientation, 0.05 meters in $l_h$, 0. 1 radians in $l_p$, 0. 1 radians in $l_t$, the resulting Euclidean distance will be 0.194, which are within this small range. For other POMDP problems, users can adjust $D_{lim}$ to ensure actions within this range have similar physical meanings with acceptable differences.

---

> ### Author Response · Authors · 2024-11-23
> **Response to Reviewer 4TPu for Submission185 (Part 2)**
>
> $\textbf{W2. Certain assumptions, such as a 100% success rate for actions in Gazebo}$...
>
> Answer: Thank you very much for this constructive comment. This is indeed a valuable question for enhancing the framework's practical applicability. You are correct that achieving a 100% success rate for actions in Gazebo is not realistic. While primitives for changing the robot's configuration and declarations can achieve near-perfect reliability, the main challenge lies in the removing actions. Unsuccessful removing execution may cause totally different challenges to our framework. Since we lack prior information about most obstacles and target objects, all estimations rely on onboard sensors. Perception errors can lead to lower success rates for removing actions. For instance, a robot may fail to pick up a detected object and instead push it down. In our framework, this scenario stops data association for the "standing object" and introduces a "newly detected object." While this increases the steps needed to find the target object, it does not usually prevent task completion. However, some failures could negatively affect overall performance.
>
> To address poor primitive reliability, we propose several approaches:
>
> $\textbf{Incorporating Action Failure in Planning:}$ Through prior experiments, we can estimate failure rates for various actions in specific scenarios and integrate these probabilities into the POMDP transition function. This would allow the planning process to account for action failures and mitigate their effects.
>
> $\textbf{Improving Primitive Success Rates:}$ Another way is to improve the successful rate of the primitives using the following possible ways:
>
> 1. $\textbf{Action Execution Monitoring}$: Introduce robust monitoring systems to detect and respond to failures during primitive action execution. This could involve incorporating sensors or feedback mechanisms to validate the outcome of each action. If a failure is detected, the system can trigger a reattempt of the removing action or take corrective measures to ensure successful execution.
>
> 2. $\textbf{Enhancing Implementation Methods}$: Enhance action implementation capabilities using more robust and advanced methods to mitigate the impact of uncertainties on primitive execution. Like, the move_it toolbox may not be robust enough in the complicated environment based on the default setting. For example, we can adjust the used configurations in move_it to improve success rates in complex environments.
>
> 3. $\textbf{Robust Perception and Sensing}$: Enhance perception and sensing capabilities to mitigate the impact of uncertainties on primitive execution. By using more robust and diverse sensory inputs, the system can better adapt to changes in the environment and make informed decisions even in the presence of imperfect or incomplete information.
>
>
> We acknowledge that robust action implementation, especially for removing actions, under any environment remains an open challenge for mobile robot manipulators with onboard sensors. Since our work primarily focuses on planning, it is difficult to address all aspects of action implementation comprehensively in this paper. Nonetheless, we aim to improve future work by relaxing the assumptions around action implementation. We have added a limitation section to explain this point clear.
>
>
> $\textbf{W3. In terms of writing quality, although the authors… }$
>
> Answer: Thank you for this comment. Our paper is built with some supporting basis, like robot control and perception, and a complete framework, which may be challenging to implement for readers unfamiliar with these operations. To address concerns about reproducibility (even though we plan to release all the code via a private GitHub repository), we have included detailed explanations in our submission to support readers in replicating our results. Additionally, we have carefully revised the paper to make it more concise, including improvements such as:
>
> (1) Replace
>
> “To determine the optimal number of clusters, we test different values of $k_i$ within a given range. For each $k_i$, the KMeans algorithm clusters data into $k_i$ groups of equal variance, minimizing the inertia or within-cluster sum-of-squares criterion $l_i$. The clustering result with the minimal $l_i$ is selected, and all clusters are enclosed within high-dimensional hyperspheres with centers $\mathcal{C}_a$ and radii $\mathcal{R}_a$.”
>
> by
>
> “We use the Elbow method Thorndike (1953) to determine the optimal number of clusters $k_i$ and all clusters are enclosed within high-dimensional hyperspheres with centers $\mathcal{C}_a$ and radii $\mathcal{R}_a$.”
>
> (2) Remove “based on the test data and the context data” in “The whole network structure without training based on the test data and the context data is shown in Appendix~F.”
>
> and so on. Please check the revised version for more details.

---

> ### Author Response · Authors · 2024-11-23
> **Response to Reviewer 4TPu for Submission185 (Part 3)**
>
> $\textbf{Q1. In Table 1, do the reward values convey anything other than}$ …
>
> Answer: Thanks for your comments. Yes the reward values in POMDP means the accumulated discounted reward, where a higher reward indicates better performance. The POMDP problem is a sequence decision making problems and an optimization problem. Its objective function is accumulated discounted reward. Hence, in our mind, as an optimization problem, it is very general to report the final objective function for comparison. We still think to report both the number of steps and objective function are better.
>
> $\textbf{Q2. Line 246 - why does it start from the middle of a sentence?}$
>
> Answer: Thank you for the comment. The explanation starting on Line 246 is intended to provide additional details about the specific functions and steps used in Algorithm 1. The phrase “where Net.sam(·) is the function that generates…” treats the algorithm as an object, with the subsequent lines elaborating on each element in the algorithm. This structure allows readers to reference the algorithm while gaining a deeper understanding of its elements. To enhance clarity, I have revised this section by replacing “Where” with:
>
> “Algorithm 1 includes functions and parameters that are detailed below:”
>
> $\textbf{Q3. Why was the time complexity not reported for any of the algorithms in Table 1? Is it efficient enough to actually use NPF-kCT as an online solver with real robots?}$
>
> Answer: All methods reported in Table 1 are limited with the same planning time at each step. In appendix I, we have a sentence about the running time limitation: “Each planning step is allotted a maximum of 60 seconds.” Recognizing the importance of this limitation, we have revised the manuscript to include this detail in the main document instead of the appendix. The revised sentence is: “For all methods reported in Table 1, the time allocated for each planning step is capped at 60 seconds.” Regarding NPF-kCT, it is an anytime solver that can be stopped based on the specified planning time, making it suitable as an online solver. However, since it relies on MCTS and executes episodes (particles) during the planning process, excessively short planning times may reduce the number of particles, potentially limiting performance. The network is pre-trained before the task and utilized during planning without breaking the online requirements. In our real robot experiments, NPF-kCT operates online with a 60-second planning time per step.

---

> > ### Comment · Reviewer_4TPu · 2024-11-27
> >
> > Thank you for the response and the detailed explanations. I will be inclined to support accepting this paper.

---

> > > ### Author Response · Authors · 2024-11-27
> > > **Thank you very much for your response.**
> > >
> > > Thank you for taking the time to review our rebuttal. We truly appreciate your effort and thoughtful engagement with our work.

---

> > > ### Author Response · Authors · 2024-11-27
> > > **Follow-Up on Review**
> > >
> > > Thank you very much again for your detailed review and kind words about our paper. We deeply appreciate your thoughtful feedback and your inclination to support its acceptance.
> > >
> > > We understand that the final decision involves both qualitative and quantitative factors. Could you consider improving the score to better reflect your positive impression? We believe this could enhance the paper's chances in the competitive review process. Sorry to trouble you.
> > >
> > > Thank you again for your support and for taking the time to review our work.

---

> ### Comment · Area_Chair_acPG · 2024-11-25
>
> Dear Reviewer,
>
> Please provide feedback to the authors before the end of the discussion period, and in case of additional concerns, give them a chance to respond.
>
> Timeline: As a reminder, the review timeline is as follows:
>
> November 26: Last day for reviewers to ask questions to authors.
>
> November 27: Last day for authors to respond to reviewers.

---

### Meta-Review · Area_Chair_acPG · 2024-12-22

**Metareview:**

This paper proposes the Neural Process Filtered k-Center Clustering Tree (NPF-kCT), an online solver for continuous POMDP-based object search. The work integrates Monte Carlo Tree Search (MCTS) with a neural process network and k-center clustering to efficiently refine the action space for mobile robots in partially observable environments. While the paper tackles an important problem for the robotics realm, it exhibits critical weaknesses that outweigh its contributions.

The reviewers raised significant concerns regarding the simplicity of the experimental scenarios, which fail to demonstrate the proposed method's utility in realistic or challenging settings. Despite incorporating simulations and limited real-world experiments, the tasks evaluated are not complex enough to substantiate the framework’s claims of generalizability and scalability. Furthermore, the methodology, while novel in its combination of existing components, relies on heavily engineered heuristics specific to object search, which reduces its appeal to the broader machine learning communit7. The paper also lacks rigorous ablation studies and sufficient comparisons with state-of-the-art methods, raising doubts about the true efficacy of the approach.

The rebuttal provided additional explanations and some minor revisions but did not adequately address the core concerns about the lack of generalization, limited complexity of the experimental settings, and insufficient evaluation. The reviewers noted that while the authors' responses clarified some points, they failed to introduce new evidence or address the need for deeper insights from experiments. As such, the paper does not meet the standards required for publication at ICLR.

**Additional Comments On Reviewer Discussion:**

During the discussion phase, the reviewers highlighted several critical weaknesses, including the limited complexity of tasks, insufficient baseline comparisons, and the lack of robust experimental validation. They also raised concerns about the narrowly engineered nature of the approach and its limited applicability to broader problems beyond object search.

The authors attempted to clarify these issues in their rebuttal, emphasizing their contributions and addressing points of confusion. However, the rebuttal primarily reiterated the existing claims without introducing substantial new experiments or insights. The scenarios presented in the paper remained overly simplistic, and the generalizability and real-world applicabilityof the method were not convincingly demonstrated. Despite some reviewers appreciating the authors’ effort to improve clarity, the core issues remained unresolved.

---

### Decision · Program_Chairs · 2025-01-22

Reject